# Grid-cell representations in mental simulation

**Jacob LS Bellmund[1,2]\*, Lorena Deuker[1,3], Tobias Navarro Schröder[1,2], Christian F Doeller[1,2]\***

[1]Donders Institute for Brain, Cognition and Behaviour, Radboud University, Nijmegen, The Netherlands; [2]Kavli Institute for Systems Neuroscience and Centre for Neural Computation, Norwegian University of Science and Technology, Trondheim, Norway; [3]Department of Neuropsychology, Institute of Cognitive Neuroscience, Ruhr University Bochum, Bochum, Germany

**Abstract** Anticipating the future is a key motif of the brain, possibly supported by mental simulation of upcoming events. Rodent single-cell recordings suggest the ability of spatially tuned cells to represent subsequent locations. Grid-like representations have been observed in the human entorhinal cortex during virtual and imagined navigation. However, hitherto it remains unknown if grid-like representations contribute to mental simulation in the absence of imagined movement. Participants imagined directions between building locations in a large-scale virtual-reality city while undergoing fMRI without re-exposure to the environment. Using multi-voxel pattern analysis, we provide evidence for representations of absolute imagined direction at a resolution of 30° in the parahippocampal gyrus, consistent with the head-direction system. Furthermore, we capitalize on the six-fold rotational symmetry of grid-cell firing to demonstrate a 60° periodic pattern-similarity structure in the entorhinal cortex. Our findings imply a role of the entorhinal grid-system in mental simulation and future thinking beyond spatial navigation.

**\*For correspondence:**
j.bellmund@donders.ru.nl (JLSB);
c.doeller@donders.ru.nl (CFD)

**Competing interests:** The authors declare that no competing interests exist.

## Introduction

Anticipation of the future is a central adaptive function of the brain and enables adequate decision-making and planning. Simulating or imagining future events and scenarios relies on a network of brain regions known to be involved in episodic memory, navigation and prediction (*Buckner, 2010*; *Byrne et al., 2007*; *Hassabis and Maguire, 2007*; *Hasselmo, 2009*; *Schacter et al., 2012*). For instance, before leaving your favorite cafe, you may picture the scenery in front of the cafe in your mind's eye to determine whether to take a left or a right turn to get home. To accomplish this you have to recall both the location of the cafe as well as the direction you are facing when leaving the building.

Electrophysiological recordings in freely moving rodents have demonstrated that positional information during navigation is represented by place cells in the hippocampus (*O'Keefe and Dostrovsky, 1971*) and grid cells in entorhinal cortex (*Hafting et al., 2005*). Place cells typically exhibit one firing field (*O'Keefe and Dostrovsky, 1971*), while grid cells are characterized by multiple firing fields arranged in a regular hexagonal pattern tessellating the entire environment (*Hafting et al., 2005*). Complementarily, directional information is carried by head direction cells, which increase their firing rate as a function of the animal's directional heading irrespective of its location (*Taube et al., 1990*; *Taube, 2007*). Intracranial recordings in patients exploring virtual-reality (VR) environments demonstrated the existence of place and grid cells in the human hippocampus and entorhinal cortex, respectively (*Ekstrom et al., 2003*; *Jacobs et al., 2010*, *2013*). A 60° directional periodicity of BOLD-signal modulations in the entorhinal cortex during virtual navigation indicates

**eLife digest** Recordings of brain activity in moving rats have found neurons that fire when the rat is at specific locations. These neurons are known as grid cells because their activity produces a grid-like pattern. A separate group of neurons, called head direction cells, represents the rat's facing direction. Functional magnetic resonance imaging (fMRI) studies that have tracked brain activity in humans as they navigate virtual environments have found similar grid-like and direction-related responses. A recent study showed grid-like responses even if the people being studied just imagined moving around an arena while lying still. Theoretical work suggests that spatially tuned cells might generally be important for our ability to imagine and simulate future events. However, it is not clear whether these location- and direction-responsive cells are active when people do not visualize themselves moving.

Bellmund et al. used fMRI to track brain activity in volunteers as they imagined different views in a virtual reality city. Before the fMRI experiment, the volunteers completed extensive training where they learned the layout of the city and the names of its buildings. Then, during the fMRI experiment, the volunteers had to imagine themselves standing in front of certain buildings and facing different directions. Crucially, they did not imagine themselves moving between these buildings.

By using representational similarity analysis, which compares patterns of brain activity, Bellmund et al. could distinguish between the directions the volunteers were imagining. Activity patterns in the parahippocampal gyrus (a brain region known to be important for navigation) were more similar when participants were imagining similar directions.

The fMRI results also show grid-like responses in a brain area called entorhinal cortex, which is known to contain grid cells. While participants were imagining, this region exhibited activity patterns with a six-fold symmetry, as Bellmund et al. predicted from the characteristic firing patterns of grid cells.

The findings presented by Bellmund et al. provide evidence that suggests that grid cells are involved in planning how to navigate, and so support previous theoretical assumptions. The computations of these cells might contribute to other kinds of thinking too, such as remembering the past or imagining future events.

that grid-like entorhinal signals can also be detected with fMRI (*Doeller et al., 2010*; *Kunz et al., 2015*; *Horner et al., 2016*).

Notably, place cell activity can also represent locations other than the one currently occupied by the animal as illustrated by activation sequences corresponding to upcoming trajectories during rest periods (*Dragoi and Tonegawa, 2011*). Intriguingly, these 'preplay' sequences preferentially represent paths leading up to motivationally relevant locations (*Ólafsdóttir et al., 2015*). These observations support the notion that prospective coding of hippocampal place cells relates to the well-established role of the human hippocampus in mental simulation and imagination (*Buckner, 2010*; *Byrne et al., 2007*; *Hassabis and Maguire, 2007*; *Hasselmo, 2009*; *Schacter et al., 2012*). Akin to firing rate increases of neurons in the human medial temporal lobe specific to the content of imagination (*Kreiman et al., 2000*), firing patterns of spatially tuned cells might be reinstated to imagine the view from a certain location during mental simulation (*Bird et al., 2012*; *Byrne et al., 2007*; *Hasselmo, 2009*). Prospective coding properties of grid cells (*De Almeida et al., 2012*; *Kropff et al., 2015*) and recent evidence for spatial coherence of grid with place cell activity during replay (*Ólafsdóttir et al., 2016*) further suggest a similar involvement of the entorhinal grid system in future anticipation and prediction. This is in line with the observation of grid-like representations during imagined movement through an environment (*Horner et al., 2016*). However, hitherto it remains unknown if grid-like representations support mental simulation independent of imagined movement, which could suggest a more general role of grid cell computations in navigational planning, future anticipation and cognition.

# Results

We combined fMRI with multi-voxel pattern analysis and VR to investigate whether the entorhinal grid system contributes to the imagination of directions from stationary viewpoints (*Figure 1a,b*). After extensive navigation training (see Materials and methods and *Figure 1—figure supplement 1*), participants were asked to imagine directions between pairs of buildings in 'Donderstown', a large-scale realistic VR city (http://www.doellerlab.com/donderstown/). In a carefully counterbalanced design, we probed the fine-grained representations of twelve equally spaced directions (see Materials and methods and *Figure 1—figure supplement 2*). Imagined directions had to be indicated (*Figure 1b*) and participants successfully performed this task (mean error 33.68° ± 19.09° SD; *Figure 1c*). Behavioral performance in the direction-imagination task was highly correlated with navigation success (r = 0.85, p<0.001) and the accuracy of direction estimates (r = 0.94, p<0.001) during

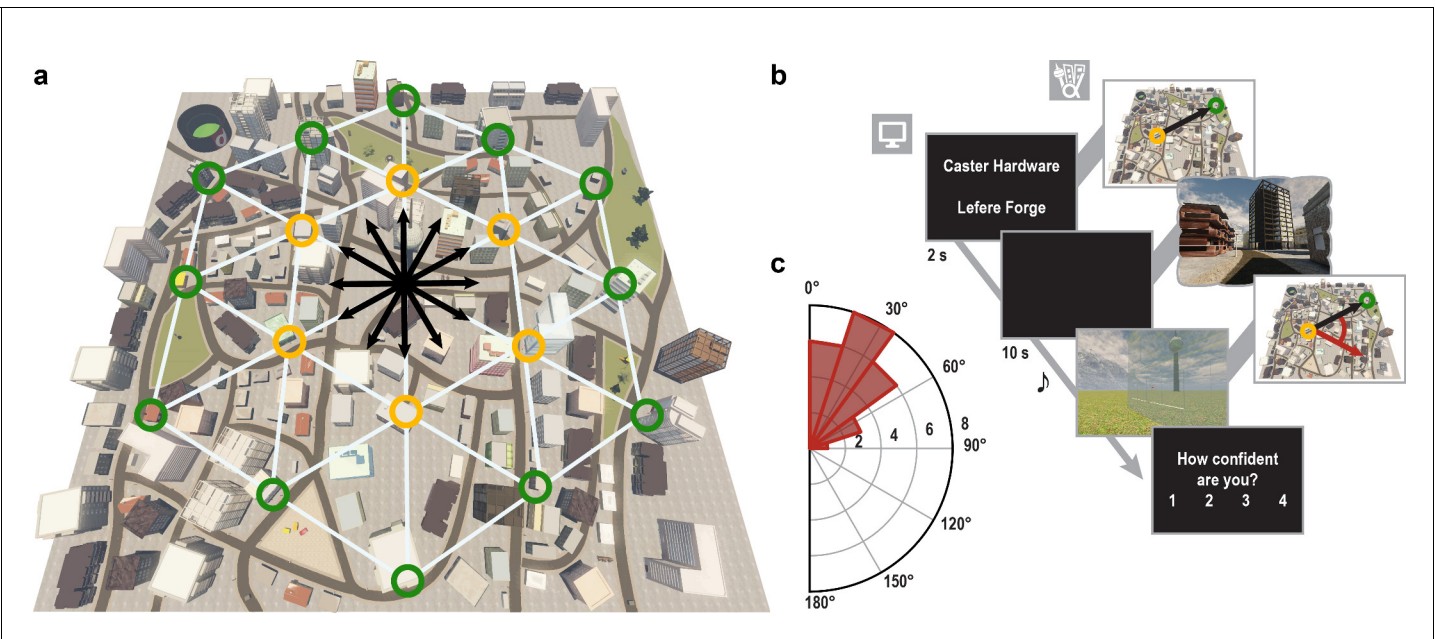

**Figure 1.** Direction-imagination task. (a) Twelve evenly spaced directions were sampled using 18 buildings distributed regularly across Donderstown. We sampled each direction (indicated by black arrows) from different start locations (yellow circles), which dissociated the directions from visual features of imagined views (*Figure 1—figure supplement 2*), and employed a counterbalancing regime ensuring equal sampling of directions and start locations throughout the experiment (see Materials and methods). Buildings marked with a green circle served as target locations only. Importantly, the regular arrangement of building locations did not correspond to the street layout and was not revealed to participants, who experienced Donderstown only from a first-person perspective (see also *Figure 1—figure supplement 1d*). (b) Trials began with a cue indicating start (top building name) and target (bottom building name) location and thereby defining the relevant direction (black arrow). During an imagination period the screen was black and participants were instructed to imagine the view they would encounter when standing in front of the start building facing the direction of the target building. An auditory signal terminated the imagination period and participants indicated the imagined direction (red arrow) in a sparse VR environment, followed by a confidence judgment. Performance was measured as the absolute angular difference between the correct and the indicated direction (red arc). Note that only the bottom row of images was presented to participants, top row for illustration only. (c) Circular histogram of average absolute angular difference between correct and indicated directions across participants (mean error 33.68° ± 19.09° SD).

The following source data and figure supplements are available for figure 1:

**Source data 1.** Average absolute angular errors.

**Figure supplement 1.** Overview of behavioral training.

**Figure supplement 2.** Sampling of directions in the imagination task.

**Figure supplement 3.** Accuracy of direction judgments during imagination task is related to behavioral performance during training and a post-scan map test.

training and performance in a post-scan map test (r = 0.95, p<0.001), indicating successful translation of the acquired representation of Donderstown to the imagination task (*Figure 1—figure supplement 3*).

The contribution of spatial representations to imagination was assessed using representational similarity analysis (*Kriegeskorte and Kievit, 2013*), which compares activation patterns across voxels to estimate neural similarity. In line with the suggested role for grid cell computations in vector navigation (*Bush et al., 2015*), we expected the grid-cell system to be involved in determining the vector comprising the direction and distance from the start to the target building in our task. Our approach focused on the direction to the target to track putative grid-cell representations during imagination by systematically comparing neural similarity of imagined directions with varying angular differences (see Materials and methods). We predicted that an involvement of the grid system in mental simulation should be reflected in a 60° periodic pattern-similarity structure in the entorhinal cortex, consistent with the hexagonal firing properties of grid cells (*Hafting et al., 2005*) and the hexadirectional fMRI signal in entorhinal cortex observed during virtual navigation (*Doeller et al., 2010*; *Horner et al., 2016*; *Kunz et al., 2015*). It is important to note that we did not rely on the building layout as an absolute reference frame in our analyses, but rather assessed pattern similarity based on the *relative* angle between the directions sampled in a trial pair, see below.

In a first step, we ascertained that absolute directional representations are detectable with our novel imagination task. We expected increased neural similarity during imagination of similar directions, consistent with previous findings of absolute directional coding during navigation in parahippocampal cortex (*Doeller et al., 2010*) and two recent studies reporting directional representations during imagination in a local reference frame in the retrosplenial complex (*Marchette et al., 2014*) and coarse representations of directions to a goal in the entorhinal/subicular region (*Chadwick et al., 2015*). However, it remains unclear whether global spatial representations are involved in human imagination in the absence of visual input.

Here, we compared pattern similarity during imagination of directions in pairs of trials sampling similar directions (angular difference $\leq$ 30°) to pairs of trials sampling dissimilar directions (*Figure 2a*) in brain regions representing facing direction (*Baumann and Mattingley, 2010*; *Chadwick et al., 2015*; *Marchette et al., 2014*; *Vass and Epstein, 2016*, *2013*). We observed the predicted one-fold symmetric pattern-similarity structure in a cluster of voxels in the left posterior parahippocampal gyrus ($T_{23}$ = 4.82, p = 0.024, FWE-corrected for multiple comparisons using small volume correction; *Figure 2b,c*; see Materials and methods). Increased pattern similarity for similar directions was not due to trial comparisons with identical building combinations (*Figure 2—figure supplement 1*). Further, this effect was not driven by the specific locations used to sample directions or the distances between these locations in Donderstown (*Figure 2—figure supplement 2*, see Materials and methods).

Having verified that we can detect directional representations in our novel imagination paradigm, we tested, in a next step, whether activation patterns during imagination follow a six-fold rotational symmetry, akin to the six-fold symmetric firing pattern of grid cells (*Hafting et al., 2005*) and the six-fold modulation of entorhinal fMRI signals during virtual (*Doeller et al., 2010*; *Kunz et al., 2015*) and imagined (*Horner et al., 2016*) navigation in humans. The rationale underlying our analysis is that activation patterns during directional imagination should exhibit the highest neural similarity for directions that are (multiples of) 60° apart from each other (see *Figure 3a–d* and *Figure 3—figure supplement 1* for details of analysis logic). Because grid cells are most abundant in the medial entorhinal cortex in rodents (*Hafting et al., 2005*), we predicted the effect to be present in posterior medial entorhinal cortex (pmEC), the likely homologue region of the rodent medial entorhinal cortex in the human brain (*Navarro Schröder et al., 2015*) (*Figure 3e*).

We observed pattern similarity increases with a 60° periodicity in the left pmEC ($T_{23}$ = 2.37, p = 0.027; one-tailed test, Bonferroni corrected for test in both hemispheres; Cohen's d = 0.48; *Figure 3f* and *Figure 3—figure supplement 2c*). The effect was further confirmed using permutation-based significance testing (pseudo $T_{23}$= 2.89, p = 0.008; see Materials and methods). A control analysis showed that the effect was not present in the anterior lateral entorhinal cortex (p>0.9; *Figure 3—figure supplement 2*; see *Figure 3—figure supplement 3* for information on signal quality in the entorhinal cortex), the human homologue of lateral EC, which does not contain grid cells (*Hafting et al., 2005*). The 60° periodicity in left pmEC was consistent across all angular differences (*Figure 3—figure supplement 4* and *Figure 3—figure supplement 5*) and the effect was not driven

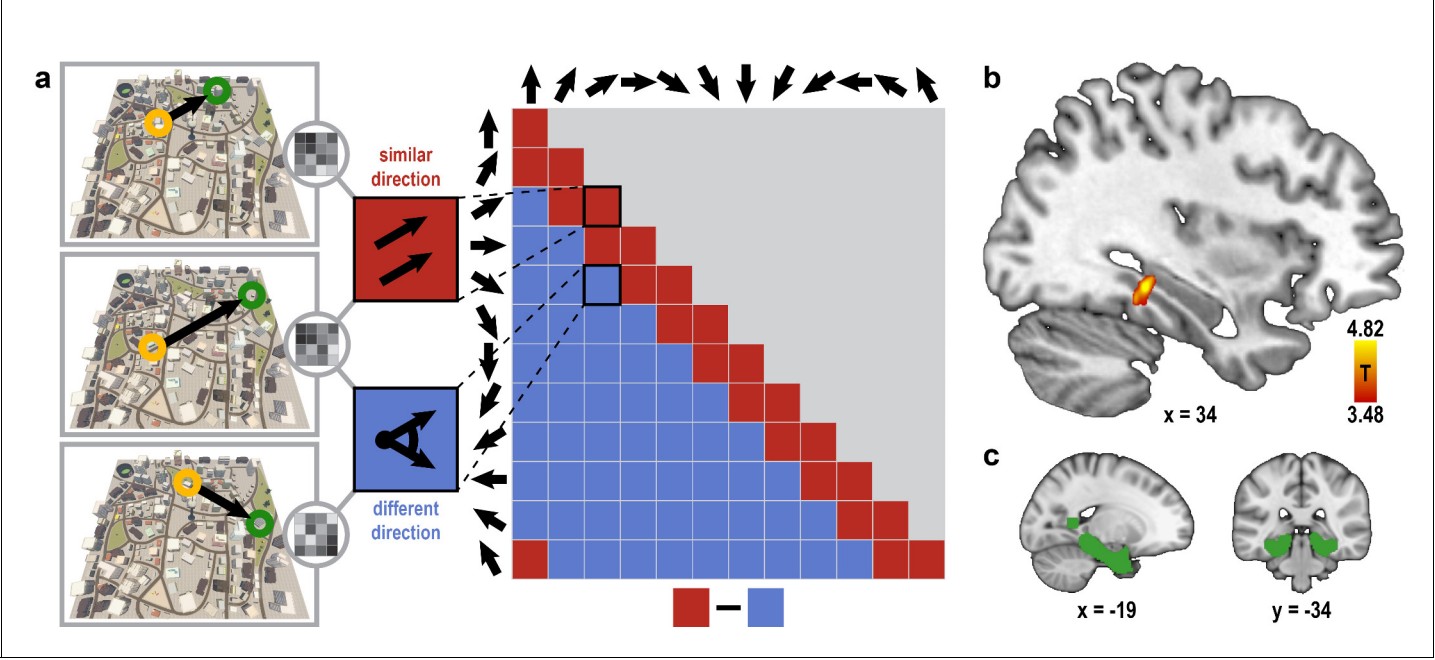

**Figure 2.** Absolute directional coding in posterior parahippocampal gyrus. (**a**) Analysis logic of the *one-fold* directional analysis for three example trials. High pattern similarity was predicted for pairs of trials sampling similar directions with a maximum angular difference of 30° (red) compared to trials sampling directions 60° or more apart (blue). Note that for illustration purposes the predicted similarity matrix is shown for comparisons across conditions, not single trials. (**b**) Searchlight results show a significant cluster of voxels in the posterior parahippocampal gyrus (peak voxel MNI coordinates: 34 -34 -10; $T_{23}$ = 4.82, p = 0.024 corrected for multiple comparisons using small-volume correction) with higher pattern similarity for trials sampling similar directions compared to trials sampling dissimilar directions. Results are shown on the structural MNI template. For display purposes, the statistical map is thresholded at p<0.001 uncorrected. (**c**) Sagittal and coronal view of the mask used to correct for multiple comparisons (see Materials and methods) displayed on the MNI template brain.

The following source data and figure supplements are available for figure 2:

**Source data 1.** Searchlight results for absolute directional coding analysis.

**Figure supplement 1.** Increased pattern similarity for similar directions after excluding trial pairs sampling a direction with the same combination of buildings.

**Figure supplement 2.** Absolute directional coding during imagination is independent of locations and distances in Donderstown.

by the specifics of our design and the VR town used. Specifically, the effect remained significant after excluding combinations of trial pairs (*Figure 3—figure supplement 6*) with the same start ($T_{23}$ = 2.39, p = 0.025) or target location ($T_{23}$ = 2.57, p = 0.017), the same combination of start and target location ($T_{23}$ = 2.45, p = 0.022) and comparisons from the same task block ($T_{23}$ = 2.08, p = 0.049). Further control analyses demonstrated that the effect was independent of the mean distance between start and target locations in a trial pair ($T_{23}$ = 2.37, p = 0.027; *Figure 3—figure supplement 7*), the difference of this distance within a pair ($T_{23}$ = 4.32, p<0.001) and the mean distance between all four buildings in a given trial pair ($T_{23}$ = 2.37, p = 0.027). Behavioral performance did not differ between the conditions ($T_{23}$ = 1.24, p = 0.227, *Figure 3—figure supplement 8*).

Furthermore, the effect was specific to a 60° modulation of pattern similarity values and there was no evidence for coding of cardinal directions in the entorhinal cortex (*Figure 3—figure supplement 9*; see also Materials and methods). Results of a whole-brain searchlight analysis confirmed the 60° periodicity of pattern similarity increases in pmEC observed in the ROI analysis ($T_{23}$ = 4.04, p = 0.046, FWE-corrected for multiple comparisons using small volume correction; *Figure 3—figure supplement 10*). A similar pattern similarity structure was observed in regions in parietal and visual cortices (*Figure 3—figure supplement 10*), which might reflect reactivation of egocentric and visual

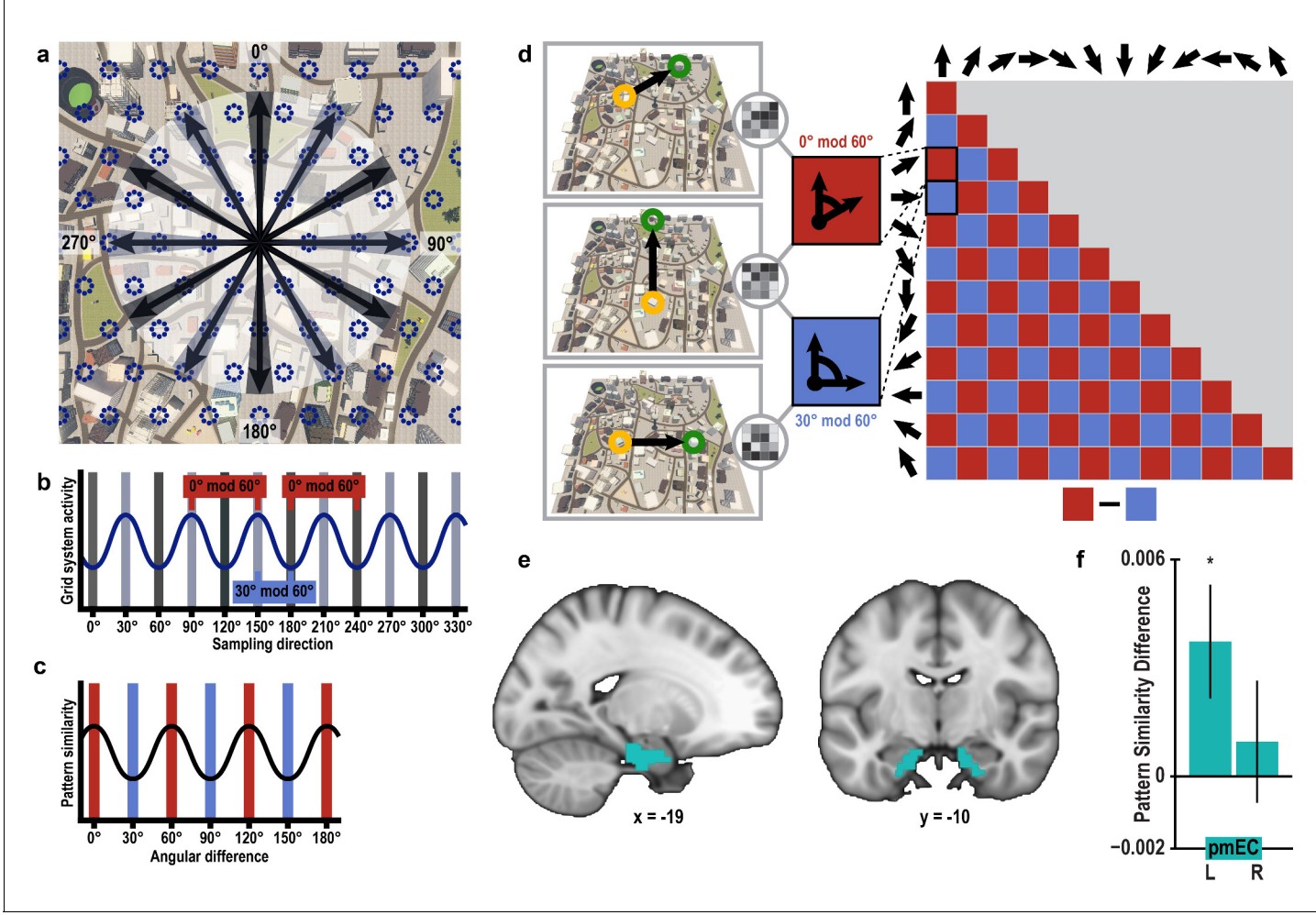

**Figure 3.** Grid-like representations during imagination. (a) Six-fold symmetric firing fields of a hypothetical grid cell (dark blue dotted circles) superimposed on an aerial view of Donderstown. Black arrows indicate the twelve sampled directions; light and dark shading highlights directions (multiples of) 60° apart. For illustration purposes, the grid orientation is aligned to the sampled directions; see *Figure 3—figure supplement 1* for a different example. (b) The firing rate of the hypothetical response of the grid-cell system as a function of direction, showing a 60° modulation. Shading displays sampling of directions and red and blue markers indicate the two conditions. Note that the oscillatory firing pattern is sampled at the same phase in the 0° modulo 60° condition, but at different phases in the 30° modulo 60° condition. (c) Based on this, we expected a 60° modulation of fMRI pattern similarity values when comparing trial pairs based on the angular *difference* of their sampled directions. Red and blue shading illustrates the two conditions. (d) Specifically, we predicted higher pattern similarity for trial pairs with a remainder of 0° (0° modulo 60° condition, red) compared to trial pairs with a remainder of 30° (30° modulo 60° condition, blue), when dividing the angular difference of the pair's sampling directions by 60°. Note that for illustration purposes the predicted similarity matrix is shown for comparisons across conditions, not single trials. (e) ROI mask for posterior medial entorhinal cortex (pmEC) from previous report (*Navarro Schröder et al., 2015*). (f) Pattern similarity difference (mean and S.E.M.) between the two conditions. The left pmEC exhibited a significant 60° modulation of pattern similarity. No significant differences in pattern similarity were observed in the right pmEC ($T_{23} = 0.57$, $p = 0.58$).

The following source data and figure supplements are available for figure 3:

**Source data 1.** Pattern similarity difference between 0° modulo 60° and 30° modulo 60° condition in left and right posterior medial entorhinal cortex.

**Figure supplement 1.** Rationale of 60° modulation analysis.

**Figure supplement 2.** Pattern similarity difference between 0° modulo 60° condition and 30° modulo 60° condition in pmEC and alEC.

**Figure supplement 3.** Signal quality in the entorhinal cortex.

**Figure supplement 4.** 60° periodicity of pattern similarity is consistent across angular differences only in left posterior medial entorhinal cortex.

*Figure 3 continued on next page*

*Figure 3 continued*

**Figure supplement 5.** Pattern similarity structure across pair-wise comparisons of trials for entorhinal ROIs.

**Figure supplement 6.** 60° modulation of pattern similarity during imagination is not driven by specifics of task design.

**Figure supplement 7.** 60° modulation of pattern similarity during imagination after controlling for distance measures.

**Figure supplement 8.** Behavioral performance for trial pairs in the 0° modulo 60° and the 30° modulo 60° condition.

**Figure supplement 9.** No evidence for representations of cardinal directions or 90° modulation of pattern similarity in the entorhinal cortex.

**Figure supplement 10.** Searchlight analysis for 60° modulation of pattern similarity during imagination.

representations associated with imagined directions, possibly modulated by entorhinal representations in line with a model of mental imagery (*Bird et al., 2012*; *Byrne et al., 2007*). Future research will need to investigate these putative interactions in more detail.

## Discussion

In sum, we report two important findings: Firstly, pattern similarity values in the parahippocampal gyrus exhibited a one-fold symmetry congruent with fine-grained representations of imagined facing direction, reflecting the role of this brain region - which has been implicated in spatial processing in the absence of visual input (*Wolbers et al., 2011*) - in representing the directional aspect of the imagined views. An alternative explanation of this effect through visual similarity of the imagined views appears unlikely due to the complex nature of the task in which each direction was sampled from multiple locations in our large-scale environment (*Figure 1—figure supplement 2*) and buildings served as cues for a wider range of sampling directions. Therefore, our finding provides the first evidence for fine-grained coding of absolute direction at an unprecedented angular resolution of 30°, consistent with the characteristics of the head direction system in rodents (*Taube et al., 1990*; *Taube, 2007*), and constitutes a three-fold increase in resolution of the directional representations observed in humans compared to previous studies (*Marchette et al., 2014*; *Chadwick et al., 2015*; *Baumann and Mattingley, 2010*; *Vass and Epstein, 2013*, *2016*; *Shine et al., 2016*). Secondly, the structure of pattern similarity in entorhinal cortex was characterized by a six-fold rotational symmetry akin to the firing properties of grid cells (*Hafting et al., 2005*). Our findings provide evidence for an involvement of grid-like representations in mental simulation in the absence of imagined movement.

Crucially, participants imagined directions from stationary viewpoints in a realistic, large-scale virtual city and were not re-exposed to the virtual town during the imagination task. Therefore, our findings provide novel evidence, complementary to a recent report (*Horner et al., 2016*) showing evidence for grid-like entorhinal processing during imagined movement through a simple virtual arena. Furthermore, in contrast, we investigated spatial processing in a large-scale, urban environment (*Stokes et al., 2015*) and, moreover focused on multi-voxel patterns. In particular, we demonstrate that this novel analysis approach, which does not rely on the estimation of the orientation of the hexadirectional signal in entorhinal cortex in an independent data set (*Doeller et al., 2010*; *Kunz et al., 2015*; *Horner et al., 2016*; *Constantinescu et al., 2016*), is sensitive to grid-like entorhinal signals by capitalizing on the six-fold symmetry of grid cell firing patterns. Contrary to the previously employed approach relying on the estimation of the orientation of the hexadirectional signal for each participant (*Doeller et al., 2010*; *Kunz et al., 2015*; *Horner et al., 2016*; *Constantinescu et al., 2016*), the individual grid orientation is not approximated using the multivariate analysis. Yet, the grid orientation might influence the strength of the grid-like entorhinal signal observed in a given participant because the sampled directions might be more or less aligned with this individual's grid orientation (*Figure 3a–c* and *Figure 3—figure supplement 1* for illustration). This needs to be taken into consideration when aiming to relate grid-like signals to behavior. However, only the multivariate approach enabled us to investigate the six-fold rotational symmetry in our

large-scale environment, in which a continuous sampling of directions as required for the estimation of the orientation of the hexadirectional signal would not have been feasible. This parsimonious approach might prove valuable for future studies investigating the role of grid-like signals in human cognition, in particular in studies with children (*Bullens et al., 2010*) or older participants (*Schuck et al., 2015*) and in clinical settings (*Hartley et al., 2007*; *Maguire et al., 2001*), where time for data acquisition is typically limited and could for instance help to further understand the putative link between the entorhinal grid system and Alzheimer's disease (*Kunz et al., 2015*).

On a theoretical level, our findings are consistent with accounts of imagination positing medial-temporal-lobe involvement in the reactivation and recombination of prior experiences (*Buckner, 2010*; *Byrne et al., 2007*; *Hassabis et al., 2007*; *Hassabis and Maguire, 2007*; *Hasselmo, 2009*; *Schacter et al., 2012*). The hippocampal formation and grid cells in particular have been implicated in path integration (*Hafting et al., 2005*; *Wolbers et al., 2007*), for which computing a homing vector based on translations from a given starting point is central (*Vickerstaff and Cheung, 2010*). Notably, the grid system is well-suited to also perform the inverse operation of calculating relative vectors between known positions in the service of navigational planning (*Bush et al., 2015*). Hence, it is plausible that the grid-cell system contributes to the calculation of vectors between start and target location during imagination (*Bird et al., 2012*; *Bush et al., 2015*; *Hasselmo, 2009*; *Horner et al., 2016*), while the head direction system (*Taube, 2007*; *Taube et al., 1990*) processes the absolute direction between the two locations (*Bird et al., 2012*; *Byrne et al., 2007*; *Hasselmo, 2009*) in our task. Our findings suggest an involvement of the entorhinal grid system in calculating vectors to target locations during navigational planning, in line with a theoretical account of vector navigation (*Bush et al., 2015*).

Functional neuroimaging can measure the firing pattern of specific cell types only indirectly (*Logothetis, 2008*). However, intracranial recordings in patients exploring virtual-reality environments demonstrated the existence of place (*Ekstrom et al., 2003*; *Jacobs et al., 2010*) and grid (*Jacobs et al., 2013*) cells in the human hippocampus and entorhinal cortex, respectively. Importantly, our results are in line with single-cell recordings in rodents that suggest a possible contribution of spatially tuned cells to future anticipation via place cell preplay of upcoming trajectories (*Dragoi and Tonegawa, 2011*) and preferential preplay of firing sequences of paths leading to motivationally relevant locations (*Ólafsdóttir et al., 2015*). Prospective coding properties of grid cells (*De Almeida et al., 2012*; *Kropff et al., 2015*) and recent evidence for grid cell replay (*Ólafsdóttir et al., 2016*) further suggest a similar involvement of the entorhinal grid system in future anticipation and prediction. By translating these ideas to human imagination, during which content-specific firing rate increases of neurons in the human medial temporal lobe have been observed (*Kreiman et al., 2000*), it is conceivable that spatially tuned cells provide the machinery for the flexible recombination of spatial and mnemonic details necessary for the construction of mental simulations (*Bird et al., 2012*; *Brown et al., 2016*; *Buckner, 2010*; *Byrne et al., 2007*; *Eichenbaum and Cohen, 2014*; *Hassabis et al., 2007*; *Hassabis and Maguire, 2007*; *Hasselmo, 2009*; *Schacter et al., 2012*) and the representation of conceptual knowledge (*Constantinescu et al., 2016*).

In concert with the recent report of grid-like processing in the entorhinal cortex during imagined navigation (*Horner et al., 2016*) our findings provide a substantial advancement for the field. Importantly, grid-like entorhinal signals during imagined navigation were observed with a similar orientation as during actual navigation through the VR environment (*Horner et al., 2016*). This finding strengthens our interpretation of the six-fold symmetric pattern similarity structure in the entorhinal cortex during imagination of directions from stationary viewpoints observed in this study as reflecting computations of the entorhinal grid system operating similarly in our realistic large-scale VR city as during navigation in smaller and simpler environments typically used in electrophysiological recording studies in rodents (*Hafting et al., 2005*) or fMRI experiments in humans (*Doeller et al., 2010*; *Horner et al., 2016*; *Kunz et al., 2015*). Importantly, the interpretation of our results as a global grid signal coding for space beyond boundaries and obstacles is in line with the report of a global grid pattern emerging with experience in rodents exploring an environment divided into two connected compartments (*Carpenter et al., 2015*).

In conclusion, we show involvement of both absolute directional parahippocampal and grid-like entorhinal signals in imagination, which provides important evidence for these representations in the absence of sensory input or imagined movement. This might suggest a more fundamental role of

spatial computations in the grid-cell system during mental simulation and possibly other forms of prospective coding and future thinking in the service of goal-directed decision-making (*Bird et al., 2012*; *Buckner, 2010*; *Byrne et al., 2007*).

# Materials and methods

## Participants

32 male participants were recruited via the online recruitment system of Radboud University Nijmegen. The study was approved by the local ethics committee (CMO Arnhem-Nijmegen, the Netherlands) and participants gave their written informed consent prior to the experiment. All participants had normal or corrected to normal vision and were compensated for their participation. Eight participants were excluded from the analysis because of motion sickness during the VR navigation training (n = 2), technical problems with the MRI scanner (n = 1) or chance-level performance during the direction imagination task (n = 5; median absolute angular error not significantly smaller than 90° as determined by Wilcoxon signed-rank test). Thus, 24 participants (age range 18–29 years, mean age 24.52 years, standard deviation 2.91 years) entered the analysis.

## Procedure

The experiment was conducted on two days and consisted of an extensive behavioral training in our virtual-reality city 'Donderstown' (http://www.doellerlab.com/donderstown/) and a direction imagination task in the MRI scanner. Additionally, participants' ability to locate the task-relevant buildings on a map was assessed. On the first day, participants learned the names and locations of 18 buildings in Donderstown in a two-hour training session in the virtual city. Before the fMRI session on the following day, participants were trained for an additional hour.

### VR city

Donderstown provides a large-scale, realistic urban environment comprising a complex layout of streets, squares and parks. Inspired by the street-map of a medievally founded German town, the layout of Donderstown resembles the irregular outline of a typical European city with curved roads. The streets of the city were not named. Participants were instructed to use the centrally positioned radio tower for orientation along with the mountain range surrounding the city. The skybox was rendered at infinity and the sun as the major cue of directional lighting was placed at the zenith above the center of the city. Thus, shadows could not serve as directional cues during any part of the experiment. 3D models of buildings for the city were created using 3ds Max (version 2014, www.autodesk.com/products/3ds-max/). The city was built and presented using the Unreal Development Kit (version 2013–07, http://udn.epicgames.com/Three/WebHome.html).

Crucially, the task-relevant buildings were placed at specific locations in Donderstown allowing us to sample twelve evenly spaced directions in the imagination task (*Figure 1a*) and to dissociate the start location from the sampled direction. Specifically, the entrance doors of these 18 buildings were approximately located at the vertices of equilateral triangles, which were arranged in a hexagon. Note that participants never saw a top-down view of the city, but only experienced the city from a first-person view and that the layout of streets and landmarks did not correspond to the regular pattern of buildings relevant for the direction imagination task. Therefore it is unlikely that participants were aware of the regular layout of task-relevant buildings (see *Figure 1—figure supplement 1d*).

### Behavioral training
#### Familiarization phase
On the first day of training, participants were familiarized with Donderstown and the controls of the computer game by freely navigating the city for ten minutes in search for a set of landmarks spread across the city (see *Figure 1—figure supplement 1a*). Participants first encountered Donderstown at a position close to the city's center facing the radio tower located in the center of the city. This position and orientation is indicated in *Figure 1—figure supplement 1a* and also served as the start location during the delivery task described below. For the familiarization phase, seven landmarks (e.g. a statue) distributed across the city were selected and participants were instructed to look for these landmarks while exploring the city. Donderstown was presented on a computer screen at a

resolution of 1024 × 768 pixels. Participants experienced Donderstown exclusively from a first-person perspective and navigated using the mouse and keyboard to control the player movements.

## Association task

Afterwards, participants learned the names of the 18 task-relevant buildings, with names randomly assigned to the buildings for each participant. We used names of unknown US-based companies consisting of two or three words. Participants were asked to remember the building-name associations during learning blocks, in which the image of a building was presented together with its name using Presentation (version 16.4, www.neurobs.com/presentation; *Figure 1—figure supplement 1b*). Learning blocks were followed by blocks of test trials during which participants had to select the building corresponding to a presented name from a set of three building images within 4 s. After each test block, participants received feedback about their performance in that block (percentage of correct answers). After reaching a learning criterion of three error-free test blocks (mean 7.00 ± 3.27 SD blocks to criterion), participants were trained in Donderstown for the rest of the 2-h training session as described below.

## Delivery task

In order to learn the building locations in the city, participants completed a delivery task (average length of navigation training 80.32 min ± 8.00 SD min), in which the goal was to find the target building whose name was presented on the screen (*Figure 1—figure supplement 1c*). To familiarize participants with the building locations, participants were guided to the buildings by following pylons until having visited each building three times. After this initial phase, participants searched for the buildings themselves, which were targeted in random order. If a participant could not find a building during the first 36 trials of this phase, a help button made a widely visible red arrow appear in the sky above the target building. We determined navigation performance by the number of buildings found after the end of the guided navigation phase. Upon finding a building, the player position was fixed for 7 s to allow time for the encoding of the building's location. Navigation was restricted to rotations to look around during this time period.

## Direction training

To emphasize the relevance of the directional relationships between buildings, participants were prompted to estimate the direction to the next target building after having found a building (*Figure 1—figure supplement 1c*). The name of the new target building was presented and participants were instructed to rotate until facing the new target. Performance on this task was measured as the absolute angular difference between the correct and the indicated direction. No explicit feedback on the direction was given, but participants were instructed to verify their direction estimates by searching for the target building in the indicated direction.

During the second training session, which took part approximately 24 hr after the first, participants completed a shorter version (1 hr) of the training procedure described above. First, participants were again trained on the building names using the *association task* described above. The learning criterion was reduced to one error-free test block (mean 1.23 ± 0.47 SD blocks to criterion). Again, participants spent the remainder of the training session performing the *delivery task* (average length of navigation training 40.44 min ± 5.09 SD min). The setup for the navigation training on the second day was identical to the previous day with the only difference that participants were guided to each building by pylons only once. Again, we measured navigational success as the number of buildings found after the guided navigation period as well as the accuracy of the direction judgments obtained.

To test the relationship of performance during training to behavioral performance in the MRI direction imagination task, the number of buildings found was aggregated across both training sessions and the accuracy of direction judgments was averaged across all estimates. The close relationship between training performance and direction estimates in the imagination task (*Figure 1—figure supplement 3*) was not due to the minor differences in the length of the navigation training between participants. Following the second training session, participants were instructed about the direction imagination task for the subsequent fMRI session and performed three practice trials in order to ensure all aspects of the task were understood correctly. All training sessions were conducted in a behavioral laboratory.

## fMRI direction imagination task

While undergoing fMRI, participants performed a novel direction imagination task. The task consisted of four blocks of 24 trials. In each trial, participants were cued with the names of two buildings and were instructed to close their eyes and imagine themselves standing in front of the start building facing the direction of the target building. To successfully imagine views from variable locations defined by the relative positions of buildings without re-exposure to Donderstown participants had to make use of their allocentric representation of the city. The angle of the vector between start and target building defined the correct direction in each trial with respect to the coordinates of Donderstown and defined the trial conditions for the representational similarity analysis (RSA) (*Kriegeskorte and Kievit, 2013*). The two building names were presented simultaneously for two seconds.

The layout of the buildings relevant for this task enabled the controlled sampling of twelve directions with an angular difference of 30° between adjacent directions (*Figure 1a*). Using the inner six buildings as start buildings for the direction imagination task allowed for the dissociation of start location and sampled direction. From each of the six start buildings ten of the twelve directions were sampled (*Figure 1—figure supplement 2*). Since the 60 unique building combinations oversampled the directions aligned with the main axes of the hexagon of buildings, we randomly excluded two trials sampling each of these directions with the constraint to exclude two building combinations per start location. The remaining 48 unique building combinations sampled each direction four times and were used for two task blocks of 24 trials. For the two blocks, trials were randomly drawn from the available combinations of start and target building with the constraints that all directions were sampled twice and each start building served as a start location four times in a block. The order of trials was randomized within blocks. For the second set of two blocks we followed the same randomization procedure.

Participants were instructed to imagine the direction between the two buildings with their eyes closed for 10 s during which the screen was black. Then they were prompted by a tone to open their eyes and indicate the imagined direction. For this behavioral response, a sparse VR environment was used (*Figure 1b*). The city was replaced by a grass plain with the only remaining cues for orientation being the radio tower, which had been located in the center of Donderstown during navigation and the mountain range surrounding the city. One of the mountains on the south-east side of the city had a summit cross on it, which had also been visible during training. In each trial, the start building was also presented on the grass plain at the position corresponding to the building's location in Donderstown. The participant was spawned at the imagined location in front of the start building with the task to indicate the previously imagined direction to the target building by rotating until a red cross presented centrally on the screen was pointing in this direction. In order to facilitate the estimation of directions to targets behind the start building, the building was turned transparent after 2 s in every trial. The accuracy of the behavioral response was measured as the angular distance between the correct direction defined by the start and target building and the response direction (*Figure 1b and c*). The magnitude of errors observed in the behavioral responses was consistent with performance levels reported in previous studies using similar tasks (*Schinazi et al., 2013*; *Zhang et al., 2014*).

Each trial ended with a confidence rating on a four-point scale and was followed by an inter-trial-interval (randomized length of 1.8, 3.6 or 5.4 s). Three trials within a block served as catch trials in which a snapshot from the VR city was presented after the imagination period, but before the onset of the response screen. This snapshot was taken from the imagined location in front of the start building and participants had to indicate within 2 s after the end of the imagination phase whether it was facing the direction of the target building or a direction different by 90°. In 50% of the catch trials, the snapshot showed the correct view. We introduced these trials in order to emphasize the relevance of vivid imagination of the relevant views. Participants responded significantly above chance for the catch trials in which they responded in time ($T_{23} = 2.70$, $p = 0.013$).

## Map test

After completion of the direction imagination task, participants' memory for the building locations was tested. In each trial, the name of a building was presented together with a map showing the outline of the city's streets. The objective of the task was to move a cursor to the location where the

building was thought to be positioned in the city. In order to facilitate orientation on the map, the landmarks participants searched during the familiarization phase were represented by symbols on the map. Participants were instructed about the identity of these symbols. Every building was to be located once. Participants' accuracy was defined as the Euclidean distance between the correct building location on the map and the location indicated by the participant.

## MRI data acquisition

Functional images were acquired at a 3 T Siemens Trio MRI system (Siemens, Erlangen, Germany) using a 3D EPI sequence with an isotropic voxel size of 2 mm and a TR of 1800 ms (TE = 25 ms, 64 slices, distance factor 50%, flip angle 15°, field of view 224 × 224 × 128 mm). Four runs corresponding to the four task blocks were collected. The duration of each run depended on the time required by the participant to complete the task block. On average, a run lasted 11.76 min (± 1.63 SD min). T1-weighted structural images were acquired with an MPRAGE sequence (TR = 2300 ms, TE = 3.03 ms, flip angle 8°, in-plane resolution = 256 × 256 mm, voxel size 1 mm isotropic).

## MRI data analysis

### Preprocessing

Preprocessing of the functional images of each of the four runs was carried out using the FSL toolbox (version 5.0.4, http://fsl.fmrib.ox.ac.uk/fsl/fslwiki/). The images were high-pass filtered (cut-off: 100 s) and motion-corrected. For the RSA each run was linearly registered to the participant's structural scan, which served as a common reference space for functional scans from all four blocks and was downsampled to an isotropic voxel size of 2 mm to correspond to the voxel size of the functional scans. A gray matter segmentation was carried out to obtain a gray matter mask for use in the analysis described below.

### Representational similarity analysis: absolute directional coding

RSA (*Kriegeskorte and Kievit, 2013*) was implemented using custom Matlab (version R2014a, www.mathworks.com/matlab/) scripts. For the preprocessed time series of every voxel, a general linear model (GLM) was calculated with the motion parameters obtained during preprocessing as predictors. The residuals of these GLMs (i.e. what could not be explained by the motion parameters) were then used for representational similarity analysis.

For this analysis, three volumes corresponding to the peak of the BOLD response of the second half of the imagination time window were averaged for each trial. Specifically, the analysis time window consisted of the volume during which the 10 s imagination period ended and the two following volumes. This time period was chosen due to the complex nature of the task, which required the participant to first retrieve the respective buildings and their locations before being able to imagine the direction between them. To rule out a potential influence of brain activity corresponding to the onset of the response screen on the analyzed volumes we conducted a whole-brain searchlight analysis (with the parameters described in detail below), which compared trial pairs with identical visual input at the onset of the response screen to pairs where this visual input differed. Since the initial response screen always showed the start building, trial pairs with the same start location would exhibit increased pattern similarity in brain areas processing visual information if the volumes analyzed contained brain activity corresponding to visual processing. No clusters showed this effect at a significance threshold of p<0.001 uncorrected and a cluster extent threshold of 10 voxels.

We performed the searchlight analyses using search spheres with a diameter of 7 voxels (1.4 mm). This approach iteratively calculates pattern similarity values for search spheres centered on the current voxel. All within brain voxels could be the center of a sphere, but within each sphere, the analysis was restricted to gray matter voxels by using the mask obtained from the structural image segmentation during preprocessing. Only search spheres including more than 30 voxels were analyzed. Pattern similarity was calculated as the Pearson correlation coefficient across all voxels within a sphere, separately for all possible pairs of trials. To test the hypothesis of increased representational similarity for similar directions, we averaged Fisher z-transformed correlation coefficients for two different conditions: Pairs of trials probing the same direction or directions with an angular difference of 30° were contrasted against pairs of trials sampling directions 60° or more apart (*Figure 2a*). We chose to investigate representations of absolute directions with a resolution of 30°

based on the firing properties of head direction cells (*Taube, 2007*), which might underlie an absolute directional code in our task, and to achieve sufficient statistical power. The difference in mean pattern similarity between these conditions was calculated for each search sphere on the single subject level.

The resulting difference images were normalized (non-linear registration) to the MNI template (2 mm resolution) using FSL and smoothed using a Gaussian kernel of 4 mm full width at half maximum. On the group level, significance was assessed by performing a one-sample T-test in SPM8 (http://www.fil.ion.ucl.ac.uk/spm/) on the difference images obtained at the subject level (cluster extent threshold 10 voxels). The resulting statistical map is displayed at 0.5 mm resolution on the MNI template in *Figure 2b*. Small volume correction was performed based on our a priori hypothesis about the brain regions involved in the representation of facing direction in humans as reported in previous studies (*Baumann and Mattingley, 2010*; *Chadwick et al., 2015*; *Doeller et al., 2010*; *Marchette et al., 2014*; *Vass and Epstein, 2013*). Therefore, a mask (*Figure 2c*) containing the retrosplenial complex, parahippocampal gyrus, entorhinal cortex and the subiculum was constructed from thresholded masks of these regions. For the subiculum and the parahippocampal gyrus masks were selected from the Jülich Histological atlas (*Eickhoff et al., 2007*) and the Harvard-Oxford Cortical Structural atlas (*Desikan et al., 2006*) distributed with FSL, respectively. The retrosplenial complex was defined based on peak voxel coordinates from a previous study in which participants imagined directions in a local reference frame (*Marchette et al., 2014*). We created a spherical mask with a radius of 6 mm around the peak voxel in the right retrosplenial complex (MNI coordinates: 14 -58 11) and added a second sphere with the same radius in the same location in the left hemisphere. Masks for the posterior medial and anterior lateral parts of the entorhinal cortex were based on a previous study (*Navarro Schröder et al., 2015*). The mask used for small volume correction is displayed at 1 mm resolution in *Figure 2c*.

## Absolute directional coding: Control analyses
### Excluding trial pairs with the same start and target location
The parahippocampus is known to be involved in the processing of visual scenes (*Epstein, 2008*). Therefore, increased pattern similarity for similar directions could potentially be driven by trial pairs in which participants were cued to imagine the identical spatial scene. We performed a searchlight analysis as described above and excluded trial pairs from the analysis, which sampled a direction with the identical combination of buildings. On the group level, we observed an effect in a very similar location in the left posterior parahippocampal gyrus (*Figure 2—figure supplement 1*).

### Exclusion of trial pairs with the same start location
To control for the possibility that increased pattern similarity was due to pairs of trials sampling directions from the same start location, we excluded these comparisons from the analysis. Specifically, we extracted the matrix containing all pair-wise correlation coefficients from the sphere centered on the peak voxel of the cluster observed in our main analysis (*Figure 2*). To this end we registered the peak voxel coordinate to each subject's structural image (2 mm resolution), which again served as the common space for the analysis of the functional images, and computed the pair-wise correlations for all possible pairs of trials in the sphere centered on this voxel (same parameters as for main searchlight analysis). After excluding all comparisons of trial pairs sampling directions from the same start location, we computed the difference in mean pattern similarity between pairs sampling similar (angular differences $\leq 30°$) and dissimilar (angular differences $\geq 60°$) directions as in our main analysis. We still observed increased pattern similarity for pairs of trials sampling similar compared to dissimilar directions ($T_{23} = 3.01$, p = 0.006; *Figure 2—figure supplement 2a*, bar I).

### Exclusion of trial pairs with the same target location
Following the same approach, we also excluded the possibility that increased pattern similarity for similar directions was driven by more frequent imagination of the same target buildings. As shown in *Figure 1 — supplement 2*, directions were sampled from the inner ring of buildings in Donderstown and mostly targeted locations towards the outside of the city. Therefore, target buildings could not be sampled from all directions. After excluding comparisons of trials targeting the same location in a

second analysis, pattern similarity remained greater for comparisons of similar directions ($T_{23}$ = 4.07, p<0.001; *Figure 2—figure supplement 2a*, bar II).

## Rationale for controlling distance-related effects

In every trial, the direction to be imagined was defined as the angle of the vector between the start and target building. In this set of control analyses we additionally considered the Euclidean distance between the start and the target location. We adopted three different approaches to combine the distance information for a given trial pair and to obtain a distance matrix with the distance for each comparison (*Figure 2—figure supplement 2b*). To test whether distances differed between conditions we computed the same contrast as in our main analysis: For each subject, we calculated the difference between the mean distance of trial pairs sampling similar directions and the mean for pairs sampling dissimilar directions and tested these differences against 0 using a one-sample t-test. The results are described below. As described above, we obtained the pair-wise similarity matrix from the sphere centered on the peak voxel of the cluster from the main analysis (*Figure 2*). In each analysis, we then computed a GLM with pairwise distances as a continuous predictor and the pairwise Fisher z-transformed correlation coefficients as the criterion for each participant. The residuals of these GLMs (i.e. pattern similarity that could not be explained by the distance predictor) were then used to compute the pattern similarity differences between trial pairs sampling directions with a maximum angular difference of 30° and pairs sampling directions with an angular difference of 60° or more. After all analyses, pattern similarity remained significantly greater for trial pairs sampling similar directions compared to trial pairs sampling dissimilar directions. The details of these analyses are described in the following.

## Controlling for mean distance of trial pairs

First, we averaged the Euclidean distances between start and target building in a trial pair. The mean Euclidean distance was greater for pairs sampling directions with a maximum angular difference of 30° ($T_{23}$ = 8.30, p<0.001). By accounting for the mean length of the two vectors, we excluded a potential influence of imagined distance to the target location. Pattern similarity remained increased for trial pairs sampling similar directions ($T_{23}$ = 3.90, p<0.001, *Figure 2—figure supplement 2c*, bar I).

## Controlling for different distances within trial pairs

Second, we considered a modulation of pattern similarity based on the difference in distance between start and target location of the trials defining a pair. Trial pairs with angular differences of maximally 30° had more different distances ($T_{23}$ = 58.23, p<0.001). After controlling for the difference in length of the two trial vectors in a pair, pattern similarity was still increased for pairs of trials sampling similar directions ($T_{23}$ = 3.61, p<0.001, *Figure 2—figure supplement 2c*, bar II).

## Controlling for the 'neighborhood distance'

Lastly, we quantified a 'neighborhood distance', which we defined as the mean of the Euclidean distances of the six vectors connecting all four buildings in a trial pair. This distance was low when both start and both target buildings were in the same part of Donderstown. These distances were smaller for pairs sampling similar directions ($T_{23}$ = −77.51, p<0.001). After controlling for the fact that buildings used to sample similar directions were located more closely together in Donderstown, pattern similarity remained increased for pairs of trials sampling similar directions ($T_{23}$ = 3.97, p<0.001, *Figure 2—figure supplement 2c*, bar III). This rules out the possibility that sampling in similar parts of Donderstown is responsible for increased pattern similarity for trial pairs sampling similar directions.

## Visual similarity of imagined views

Based on the well-established role of the parahippocampal cortex in scene processing, increased pattern similarity for similar directions in this region could potentially be driven by increased visual similarity of the scenes imagined by the participants. Even though this appears unlikely due to the sampling of each direction from multiple locations (*Figure 1—figure supplement 2*) we compared the visual similarity of the scenes to be imagined based on a set of visual features (*Milivojevic et al., 2015*). For each trial, the view from the start position facing the direction of the target building was captured at a resolution of 1024 × 768 pixels. Note that participants did not see these images

during the task, but were instructed to imagine the views. Five statistics were used to quantify visual features of each image: By computing a two-dimensional discrete Fast-Fourier-Transform we obtained magnitude and phase metrics of the images. Next, the red, green and blue values for each image were converted to CIELAB color space. This color space provides information about the luminance and two color opponent dimensions (red-green and yellow-blue) corresponding to the cone responses of the human retina. Each of the five image statistics was vectorized and Pearson correlations were computed to quantify visual similarity across pairs of images.

We compared visual similarity for pairs of trials sampling similar directions (treating angular differences of 30° or less as similar akin to the absolute directional coding analysis) from different start positions. For each participant, we performed a two-sample t-test on the Fisher z-transformed correlation coefficients of each image dimension. While we did not observe significant differences for the magnitude information or the color dimensions (all Bonferroni-adjusted p>0.2), phase and luminance information of imagined scenes were significantly less similar for comparisons of trials sampling similar directions compared to pairs sampling dissimilar directions in a subset of participants. Specifically, Fisher z-transformed correlations were significantly lower (corrected for 24 comparisons) for image pairs from similar direction trials for twelve participants when considering the phase information (minimum significant $T_{3838} = -3.25$; maximum significant $T_{3838} = -4.42$) and for six participants when considering luminance information (minimum significant $T_{3838} = -3.11$; maximum significant $T_{3838} = -4.70$). Based on these results it seems highly unlikely that differences in visual similarity of the scenes to be imagined drive the observed increased representational similarity of trial pairs sampling similar directions in the parahippocampal gyrus.

## Representational similarity analysis: six-fold symmetry

In rodents, grid cells are typically found in the medial entorhinal cortex (*Hafting et al., 2005*). Recently, two reports suggested the posterior medial part of the human entorhinal cortex as the human homologue of the rodent medial entorhinal cortex based on local and global connectivity patterns (*Maass et al., 2015*; *Navarro Schröder et al., 2015*). Therefore, we expected an influence of grid-like representations on imagination only in the posterior medial entorhinal cortex (pmEC). The anterior lateral part of the entorhinal cortex (alEC) served as a control region. ROI masks for the left and right pmEC and alEC from our previous report (*Navarro Schröder et al., 2015*) were first registered to the MNI template at a resolution of 2 mm using FSL (1 mm resolution for display in *Figure 3e* and *Figure 3—figure supplement 2a*). For each participant, the masks were then warped to the anatomical scan with a resolution of 2 mm, which again served as the reference space for the different runs.

The analysis was performed on the same preprocessed and motion-corrected volumes as the one-fold symmetry analysis. Again, RSA was performed on the residuals of a GLM with the motion parameters as predictors. For each ROI, we calculated pattern similarity for all pairs of trials and defined two conditions to compare Fisher z-transformed Pearson correlations coefficients. We hypothesized that a possible influence of grid-cell representations on imagination should be reflected in a 60° periodicity of pmEC activity patterns (*Figure 3a,b* and *Figure 3—figure supplement 1a,b*) analogously to the 60° modulation of the amplitude of the BOLD signal observed during navigation (*Doeller et al., 2010*). Extending this previous method, we pursued a more parsimonious and more powerful multivariate approach, which does not rely on the univariate estimation of a putative grid orientation on an independent dataset (which is only feasible with exhaustive sampling of directions during navigation [*Doeller et al., 2010*]). Here, pattern similarity in pmEC was expected to be modulated by the angular difference of the directions sampled in a pair of trials (*Figure 3c* and *Figure 3—figure supplement 1c*). To test this hypothesis two conditions were defined based on the modulus 60° of the angular difference between the sampled directions in a trial pair (*Figure 3d*). This remainder of modulus 60° could either be 0° or 30° due to the regular sampling of directions in our paradigm. We calculated the difference in mean pattern similarity between the 0° modulo 60° and the 30° modulo 60° condition for each subject. On the group level, we tested for increased pattern similarity in the 0° modulo 60° condition by comparing these pattern similarity difference values against 0 using a one-tailed one-sample T-test. Reported p-values are Bonferroni corrected for separate comparisons in left and right pmEC. To visualize whether the significant difference observed between the conditions in pmEC reflected a consistent 60° modulation of

pattern similarity across angular differences, we plotted pattern similarity as a function of angular difference between trials pairs (*Figure 3—figure supplement 4*).

## Permutation-based significance testing

To further assess the increased pattern similarity for trials from the 0° modulo 60° and the 30° modulo 60° condition in left pmEC, we additionally performed a permutation-based non-parametric significance test. We computed the probability of the observed pattern similarity difference between the conditions under a permutation-based null distribution for each participant. Null distributions were obtained by shuffling the trial labels and analyzing the resulting correlation matrix for each of 10,000 permutations. The resulting p-value was converted to a z-statistic (allowing both positive and negative values). On the group level, we used a one-sample t-test (non-parametric with 10,000 permutations) to test these z-statistics against 0.

## Grid-like entorhinal signals during imagination: Control analyses

As can be seen in a *Figure 1a*, the buildings relevant for the direction imagination task were distributed in a hexagonal pattern across Donderstown. A series of control analyses described in detail below suggests that the 60° modulation of pattern similarity in pmEC was not imposed by the layout of Donderstown and the specifics of the task design (*Figure 3—figure supplement 6* and *Figure 3—figure supplement 7*).

### Exclusion of trial pairs with the same start location

The number of comparisons of trials with the same start location was different between the 0° modulo 60° and the 30° modulo 60° condition. Thus, the 60° modulation of pattern similarity could have potentially been driven by the number of trial pairs in which the same start building had to be imagined. To account for this possibility, we excluded all pairs consisting of trials with the same start location in a first control analyses (*Figure 3—figure supplement 6*, bar I).

### Exclusion of trial pairs with the same target location

With respect to the target locations, a similar aspect was considered. Due to the sampling of directions from the start locations near the center of Donderstown, buildings could only serve as targets for trials sampling a limited range of directions. Since trial pairs with the same target location were more likely to belong to the 0° modulo 60° condition, we excluded comparisons of trials with the same target location to rule out that the 60° modulation of pattern similarity in pmEC was driven by more frequent imagination of the same target building (*Figure 3—figure supplement 6*, bar II).

### Exclusion of trial pairs with the same combination of start and target location

A third control analysis excluded pairs of trials with the same start and target building combination to ensure that imagination of identical scenes did not drive the effect (*Figure 3—figure supplement 6*, bar III).

### Exclusion of trial pairs from the same MRI run

To rule out a potential influence of temporal autocorrelation, we restricted the analysis to pairs of trials from different task blocks. The effect also remained significant when excluding all comparisons of trials from the same block (*Figure 3—figure supplement 6*, bar IV).

### Exclusion of trial pairs with target locations in inner ring of buildings

In some trials, directions were sampled using target buildings located on the inner ring of buildings (see *Figure 1—figure supplement 2*). Average angular error was lower for trials targeting a building on the inner compared to the outer ring ($T_{23} = -5.29$, p<0.001). Pairs in which both trials sampled directions using target locations on the inner ring made up 12.23% of comparisons in the 0° modulo 60° condition and did not occur in the 30° modulo 60° condition. When excluding these comparisons from the analysis, pattern similarity remained increased for the 0° modulo 60° condition compared to the 30° modulo 60° condition (*Figure 3—figure supplement 6*, bar V).

## Rationale for controlling distance-related effects

In a second set of control analyses we additionally considered the Euclidean distance between the start and the target location as described above for the absolute directional coding effect. Again we adopted three different approaches to combine the distance information for a given trial pair (*Figure 2—figure supplement 2a*) and computed a pairwise distance matrix for each measure. We assessed whether a distance measure differed between the 0° modulo 60° and the 30° modulo 60° condition by testing the difference between the mean distance values of the two conditions against 0 using a one-sample t-test, analogous to our approach in the pattern similarity analysis. The results of these tests are reported in the following paragraphs. In each control analysis, we then computed a GLM with a continuous distance predictor and pairwise Fisher z-transformed correlation coefficients as the criterion. The residuals of these GLMs (i.e. pattern similarity that could not be explained by the distance predictor) were then used to compute the pattern similarity differences between the 0° modulo 60° and the 30° modulo 60° condition (*Figure 3—figure supplement 7*). After all analyses, pattern similarity remained significantly greater for trial pairs in the 0° modulo 60° condition in the left pmEC. The effect also remained significant when using binary (high vs. low) distance predictors (all $T_{23} > 2.44$, all $p<0.03$).

## Controlling for mean distance of trial pairs

First, we averaged the lengths of the two vectors. The mean Euclidean distance was higher for pairs in the 0° modulo 60° condition ($T_{23} = 5.25$, $p<0.001$). By accounting for the mean length of the two vectors, we exclude a potential influence of imagined distance to the target location on pattern similarity (*Figure 3—figure supplement 7*, bar I).

## Controlling for different distances within trial pairs

A second possibility could be a modulation of pattern similarity based on the difference in Euclidean distance of the trials defining a pair. Trial pairs from the 0° modulo 60° condition had more similar distances ($T_{23} = -57.02$, $p<0.001$). Potentially, pattern similarity could be increased for trials of similar length. Therefore, we used the difference in length of the two trial vectors in a pair as a predictor (*Figure 3—figure supplement 7*, bar II).

## Controlling for the 'neighborhood distance'

Lastly, we quantified the distance of the six vectors connecting all four buildings in a trial pair. This resulted in a 'neighborhood distance', which was low when both start and both target buildings were in the same area of Donderstown. These distances were larger for pairs in the 0° modulo 60° condition ($T_{23} = 54.20$, $p<0.001$). We attempted to control for effects of imagining directions in specific parts of Donderstown with this analysis (*Figure 3—figure supplement 7*, bar III).

## Representations of cardinal directions

To examine whether potential representations of the cardinal directions were detectable in the pattern similarity structure exhibited by the pmEC during the direction imagination task, we compared pattern similarity values of pairs where both trials sampled a cardinal direction against pairs where this was not the case (*Figure 3—figure supplement 9a*). The cardinal directions were defined based on the orientation of Donderstown as displayed for example in *Figure 1a*. Pattern similarity was not increased for cardinal direction pairs in either pmEC (left: $T_{23} = -0.136$, $p = 0.893$; right: $T_{23} = -0.449$, $p = 0.658$, *Figure 3—figure supplement 9b*) or alEC (left: $T_{23} = 0.266$, $p = 0.793$; right: $T_{23} = 0.530$, $p = 0.601$).

To ascertain that trial pairs sampling directions using buildings located on the same street running along the North-South axis of Donderstown (based on its orientation for example in *Figure 1a*), which could possibly bias participants' representations of the city due to a number of streets roughly aligned with it, were not driving our effects, we excluded all comparisons of pairs in which both trials sampled directions along this street from the analysis (see *Figure 1 —figure supplement 2*). Pattern similarity was significantly greater for pairs of trials sampling similar directions compared to pairs with larger angular differences ($T_{23} = 4.00$, $p<0.001$) in the peak voxel of the parahippocampal cluster from the main absolute directional coding analysis (*Figure 2*), indicating that absolute directional coding was independent of pairs of trials sampling directions along the same street running along the North-South axis. Similarly, when examining the 60° modulation of pattern similarity values in left

pmEC after excluding all trial pairs sampling along this street, the difference between the conditions remained significant ($T_{23}$ = 2.40, p = 0.025).

A number of considerations further make an over-representation of the North-South axis due to directions sampled on a street running along this axis unlikely. Participants' performance did not differ between trials sampling along the North-South axis compared to other directions. We did not observe differences in accuracy (mean ± standard deviation (SD) for average absolute angular errors of trials sampling North-South 35.9° ± 24.5° vs. 33.2° ± 18.5° for other directions; means not significantly different: $T_{23}$ = 1.06, p = 0.300) or consistency (mean ± SD of individual standard deviations for absolute angular errors for North-South trials 31.2° ± 19.5° vs. 30.1° ± 13.5° for other directions; standard deviations not significantly different: $T_{23}$ = 0.54, p = 0.596) of participants' direction estimates. Furthermore, participants were recruited from the student body of a Dutch university and were therefore most familiar with irregular European street layouts (unlike the rectangular grid network common in the US), in which the use of cardinal directions is less common because most streets are curved and do not follow the cardinal directions. Importantly, even if participants preferentially represented directions along the North-South axis, which is unlikely based on the reasons laid out above, the population response of entorhinal grid cells would still remain the most likely explanation for the observed 60° modulation of pattern similarity values in pmEC, which could not be explained by the presence of one cardinal axis.

## Four-fold symmetry

To further corroborate the specificity of the 60° modulation observed in pmEC, we conducted a control analysis testing for a 90° modulation of pattern similarity values. A four-fold similarity pattern would be inconsistent with grid-cell like representations (*Doeller et al., 2010*; *Kunz et al., 2015*). We compared pattern similarity as function of angular difference for trial pairs with a 0° modulo 90° against trials pairs with 30° or 60° modulo 90° (*Figure 3—figure supplement 9c*). There was no significant 90° modulation of pattern similarity in either pmEC (left: $T_{23}$ = −0.48, p = 0.637; right: $T_{23}$ = −1.81, p = 0.084, *Figure 3—figure supplement 9d*) or alEC (left: $T_{23}$ = −0.83, p = 0.413; right: $T_{23}$ = 0.50, p = 0.618).

## Estimation of signal-to-noise ratio in the entorhinal cortex

In order to compare signal quality in the entorhinal ROIs we calculated the temporal signal-to-noise ratio (tSNR). We quantified tSNR as the mean signal within a region divided by the standard deviation of this signal over time. A two-way repeated measures ANOVA with the factors region (pmEC vs. alEC) and hemisphere (left vs. right) revealed no significant differences in tSNR (no main effect of region ($F_{1,23}$ = 0.60, p = 0.448), no main effect of hemisphere ($F_{1,23}$ = 0.00, p = 0.953) and no interaction ($F_{1,23}$ = 0.97, p = 0.336); *Figure 3—figure supplement 3*).

## Whole-brain searchlight analyses

Our main analysis and subsequent control analyses described above focused on the head direction network and pmEC, respectively, based on our specific a priori hypotheses and previous studies (*Baumann and Mattingley, 2010*; *Chadwick et al., 2015*; *Doeller et al., 2010*; *Marchette et al., 2014*; *Vass and Epstein, 2016*, *2013*). In particular, our pmEC ROI was motivated anatomically by the presence of grid cells in the rodent medial entorhinal cortex (*Hafting et al., 2005*) and by the functional identification of its human homologue region in our previous work (*Navarro Schröder et al., 2015*). Due to our complex novel imagination paradigm, set in a realistic large-scale VR city, we opted to further explore our data. We performed exploratory whole-brain searchlight analyses with the same parameters as reported above to test whether additional brain regions might display absolute directional coding and a 60° modulation in their pattern similarity structure, respectively. No brain regions other than the parahippocampal gyrus were observed for the absolute directional effect at a significance threshold of p<0.001 uncorrected and a voxel extent threshold of ten voxels. Results for the six-fold symmetry whole-brain searchlight analysis are shown in *Figure 3—figure supplement 10*. We did not perform searchlight analyses within pmEC or alEC due to the small size of these entorhinal subregions.

## Exploration of behavioral data

We investigated whether a 60° modulation of absolute angular error values was also present in the behavioral data. We multiplied angular error values to combine performance for all pairs of trials and compared the resulting values as a function of angular difference of the directions sampled in a pair. Analogous to the 60° modulation analysis, we calculated the difference between combined error values in the 0° modulo 60° and the 30° modulo 60° condition for each participant on the first level and tested for a difference on the second level using a one-sample t-test. The combined error values were not different between the conditions ($T_{23} = 1.24$, p = 0.227, *Figure 3—figure supplement 8*).

We further explored the behavioral data obtained during the direction imagination task. To assess whether the distance between the start and target building of a trial was predictive of the angular error of that trial, we calculated Pearson correlations between the absolute angular error and the distance from start to target building for each participant. This relationship did not reach statistical significance in any of the participants (mean r = −0.08 ± 0.09 standard deviation, range −0.24–0.10; all Bonferroni-adjusted p>0.454).

## Acknowledgements

This work was supported by the European Research Council (ERC-StG RECONTEXT 261177) and the Netherlands Organisation for Scientific Research (NWO-Vidi 452-12-009). The authors would like to thank S Collin and B Milivojevic for comments on the manuscript and A Backus for helpful discussions.

## Additional information

### Funding

| Funder | Grant reference number | Author |
| --- | --- | --- |
| European Research Council | ERC-StG RECONTEXT 261177 | Tobias Navarro Schröder Christian F Doeller |
| Nederlandse Organisatie voor Wetenschappelijk Onderzoek | NWO-Vidi 452-12-009 | Jacob LS Bellmund Lorena Deuker Christian F Doeller |

The funders had no role in study design, data collection and interpretation, or the decision to submit the work for publication.

### Author contributions

JLSB, Conception and design, Acquisition of data, Analysis and interpretation of data, Drafting or revising the article; LD, Conception and design, Acquisition of data, Analysis and interpretation of data; TNS, Conception and design, Acquisition of data; CFD, Conception and design, Analysis and interpretation of data, Drafting or revising the article

### Author ORCIDs

Jacob LS Bellmund, http://orcid.org/0000-0002-2098-4487
Lorena Deuker, http://orcid.org/0000-0002-4939-5862
Tobias Navarro Schröder, http://orcid.org/0000-0001-6498-1846

### Ethics

Human subjects: The study was approved by the local ethics committee (CMO Arnhem-Nijmegen, The Netherlands) and participants gave their written informed consent prior to the experiment.

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
