## [Decision Letter]

Thank you for submitting your article "Grid-cell representations in mental simulation" for consideration by *eLife*. Your article has been favorably evaluated by three reviewers, one of whom, Timothy EJ Behrens (Reviewer #3) also served as Reviewing and Senior Editor. Joshua Jacobs (Reviewer #2) has agreed to reveal his identity.

The reviewers have discussed the reviews with one another and the Reviewing Editor has drafted this decision to help you prepare a revised submission.

Summary:

Bellmund and colleagues' study investigated the presence of a grid-like signal (a fMRI proxy to rodent grid cells) in the human entorhinal cortex during mental simulations. Namely, the authors aimed to determine whether one could detect a grid representation in the fMRI BOLD signal while study participants carried out a spatial task requiring them to imagine the direction one would need to face to navigate from one place in a virtual environment to another. This is an important area of research as grid cells, in the rodent brain, are thought to underlie navigation and support general mnemonic processes. Moreover, recent theoretical accounts suggest they may be in important for planning future navigation (Bush, Barry et al. (2015)). As such, finding grid representations during navigational simulations would support claims and provide more general insight into how the brain prepares for future behaviour. The study is a theoretically important extension of existing work in this area, suggesting that grid cells are important for general types of cognition, which do not align to current physical movement. The experiment is well designed and introduces new methodology into the field. The inclusion of the many control analyses and visualisations of the constituent parts of the data are helpful towards supporting the reported findings.

Essential revisions:

1) As the authors are aware, there is overlap between this study and a concurrently run study recently published in Current Biology (Horner et al), which reports a similar signal during imagined physical movement. The authors acknowledge this in the paper. The reviewers clearly support publication of the current study. First, we believe it contains substantial novelties over the Horner study (explained shortly), and second we appreciate the replication of such an interesting signal in two concurrently run studies. All reviewers, however, were unsure about the authors attempt to disambiguate the two studies by claiming that imagining directions to goals (rather than imagining physical movement) took the study out of the domain of spatial navigation. We agree that it is a different phenomenon. We think that the current study widens the scope of grid-like activity to the domain of navigational planning of the sort that is predicted theoretically in (Bush et al. Neuron 2015). Because the imagined directions could not be directly enacted (because many of them passed through buildings), we also think that the current result implies a global grid-like signal of the sort investigated in (Carpenter et al. Current biology.), but for imagined navigation. We also think that the novel finding of head-direction specific activity in the current manuscript is a potentially important one. We think that the authors could focus on these strengths of the current manuscript, and do not need to rely on the claims about non-spatial processes that are more difficult to justify in a completely spatial experiment.

Technical questions:

The authors show a head direction-like signal in the parahippocampal gyrus during mental simulations. That is, using a pattern similarity approach they analyse voxels in the parahippocampal gyrus during different imagined directions and find similar (<30°) imagined directions show similar patterns of activity in the parahippocampal gyrus. Although the authors perform controls to rule out visual similarity confounds, I was not convinced they have shown this effect was independent of location in the virtual environment. That is, did comparing trial pairs where a subject imagined the same direction but in two different parts of the environment show the same pattern similarity as comparing trials where the imagined direction happened in the same part (or similar part) of the virtual environment? To convincingly show their result reflects a head direction signal they need to control for location in the environment. They could perform this control analysis by correlating pattern similarity values against distance between the places for a given trial pair. Alternatively, they could input both direction and location into the same GLM, to see if there is an independent effect of direction once direction has been controlled for.

The authors show a grid-like representation during imagined directions, such that pattern similarity was higher when comparing trials 60° apart than 30° apart. The authors perform various control analyses to confirm the robustness of this result. The analysis ruling out the effect of cardinal directions is not reported in the main text. In this analysis they exclude trials that sample the north-south axis of the environment, doing this makes their main effect non-significant. They then use a different exclusion criterion, which leaves the main effect significant. First of all, the difference in exclusion criteria is not clear. It would be useful if the authors clarified this point. Moreover, given how much text is dedicated to this result in the method it should be included in the main text. Permitting the method that maintains the significance of their main effect is reasonable and fair, they could just report that result. Finally, it would be useful to clarify why so many trials seem to sample the north-south axis, as the author suggest lack of statistical power is what makes their main effect non-significant when you exclude these trials.

The new technique is interesting in that it removes the requirement to estimate the grid angle, but as far as we can see, it does not remove it completely because the sensitivity of the technique depends on the subject-by-subject grid angle (if the grid angle is 15° or 45° there is no sensitivity at all, but if the grid angle is 0 or 30 the sensitivity is maximal). This is due to the minimal sampling of directions. Essentially, then, the technique is similar to fitting one of sin(6θ) or cos(6θ) and ignoring the other one and taking the absolute values of the betas. Obviously there is a sensitivity gain from pooling over voxels, which may regain much of the sensitivity loss. This is obviously ok, if you find a significant effect, but these issues are not well presented in the manuscript. A naive reader might easily think that there is some magic whereby there is no longer any requirement to know the grid angle. This should be clarified.

Related to the comment above, the technique presumably makes it harder to examine individual differences as the sensitivity of the technique is so dependent on the grid angle, which will differ between subjects. Is this one potential reason why the authors do not report brain-behaviour correlations? Again, the authors should discuss.

In Figure 3—figure supplement 7, the authors write "We considered the distances between start and target locations in a trial pair by controlling for three distance measures, which differed between the 0 modulo 60 and the 30 modulo 60 condition". In what way? Were the 0 modulo 60 distances longer than 30 modulo 60? It is important to document this even if you can still find the hexagonal results after working with the residuals.

More broadly, the angle selection procedure is a very clever way to increase the sensitivity of the experiment, but it raises concerns that there might be differences on average between the 30° differences and the 60° differences. Is there a way of alleviating these concerns by graphing the differences according to various metrics such as how often they include the inner vs outer ring? Such as average behavioural performance included in the two different types of trial?

There is also a concern with pattern similarity techniques of autocorrelation. We have thought about it but the various combinations are so mind-bendingly complex that it is hard to know whether this will be true. Are there some trials that are more often included in the 60° pairs than the 30° pairs? If there are more or less of these than their complements (trials that occur mostly in the 30° pairs), is it possible that they can cause some statistical autocorrelation in the question of interest. We think this is unlikely, but would be interested to hear your thoughts. We are aware that this is a weakly specified criticism. If no analyses come to mind, some comment to the reviewers would suffice.

---

## [Author Response]

Essential revisions:

*1) As the authors are aware, there is overlap between this study and a concurrently run study recently published in Current Biology (Horner et al), which reports a similar signal during imagined physical movement. The authors acknowledge this in the paper. The reviewers clearly support publication of the current study. First, we believe it contains substantial novelties over the Horner study (explained shortly), and second we appreciate the replication of such an interesting signal in two concurrently run studies. All reviewers, however, were unsure about the authors attempt to disambiguate the two studies by claiming that imagining directions to goals (rather than imagining physical movement) took the study out of the domain of spatial navigation. We agree that it is a different phenomenon. We think that the current study widens the scope of grid-like activity to the domain of navigational planning of the sort that is predicted theoretically in (Bush et al. Neuron 2015). Because the imagined directions could not be directly enacted (because many of them passed through buildings), we also think that the current result implies a global grid-like signal of the sort investigated in (Carpenter et al. Current biology.), but for imagined navigation. We also think that the novel finding of head-direction specific activity in the current manuscript is a potentially important one. We think that the authors could focus on these strengths of the current manuscript, and do not need to rely on the claims about non-spatial processes that are more difficult to justify in a completely spatial experiment.*

We appreciate the careful consideration of our findings with respect to the existing literature. We agree that the imagination of directions in our VR city is a task inherently spatial in nature. Therefore, we follow the suggestions by the reviewers to emphasize that we observed grid-like entorhinal signals in the absence of virtual or imagined movement. Further, we discuss our findings in more detail with respect to the proposed functions of grid-cell computations in vector navigation (Bush et al., Neuron, 2015) and the report of a global grid signal in a multi-compartment environment (Carpenter et al., Current Biology, 2015). Indeed, we believe that our results can be interpreted as a global grid-like signal independent of obstacles in the environment.

The revised sections of the manuscript now read as follows:

Abstract:

“However, hitherto it remains unknown if grid-like representations contribute to mental simulation in the absence of imagined movement.”

Introduction:

“However, hitherto it remains unknown if grid-like representations support mental simulation independent of imagined movement, which could suggest a more general role of grid cell computations in navigational planning, future anticipation and cognition.”

Discussion:

“Our findings provide evidence for an involvement of grid-like representations in mental simulation in the absence of imagined movement.”

Discussion:

“The hippocampal formation and grid cells in particular have been implicated in path integration (Hafting et al., 2005; Wolbers et al., 2007), a central aspect of which is computing a homing vector based on translations from a given starting point (Vickerstaff and Cheung, 2010). […] Our findings suggest an involvement of the entorhinal grid system in calculating vectors to target locations during navigational planning, in line with a theoretical account of vector navigation (Bush et al., 2015).”

Discussion:

“Importantly, the interpretation of our results as a global grid signal coding for space beyond boundaries and obstacles is in line with the report of a global grid pattern emerging with experience in rodents exploring an environment divided into two connected compartments (Carpenter et al., 2015).”

Discussion:

“In conclusion, we show involvement of both absolute directional parahippocampal and grid- like entorhinal signals in imagination, which provides important evidence for these representations in the absence of sensory input or imagined movement.”

*Technical questions:*

*The authors show a head direction-like signal in the parahippocampal gyrus during mental simulations. That is, using a pattern similarity approach they analyse voxels in the parahippocampal gyrus during different imagined directions and find similar (<30*°*) imagined directions show similar patterns of activity in the parahippocampal gyrus. Although the authors perform controls to rule out visual similarity confounds, I was not convinced they have shown this effect was independent of location in the virtual environment. That is, did comparing trial pairs where a subject imagined the same direction but in two different parts of the environment show the same pattern similarity as comparing trials where the imagined direction happened in the same part (or similar part) of the virtual environment? To convincingly show their result reflects a head direction signal they need to control for location in the environment. They could perform this control analysis by correlating pattern similarity values against distance between the places for a given trial pair. Alternatively, they could input both direction and location into the same GLM, to see if there is an independent effect of direction once direction has been controlled for.*

We understand the concern raised by the reviewers that coding for locations or parts of the virtual city might play a role for the absolute directional signal observed in the parahippocampal gyrus (PHG). Following the reviewers’ advice, we conducted two sets of control analyses to demonstrate that the observed head direction-like signal is independent of location in Donderstown and cannot be explained by the distances between the buildings used to sample directions. To demonstrate that absolute directional coding in the parahippocampal gyrus was not driven by building locations we excluded (I) comparisons of trials sampling directions from the same start location and (II) comparisons of trials using the same target building. Specifically, we obtained the pair-wise similarity matrix from the peak voxel of the absolute directional coding effect of our main analysis (Figure 2) and calculated the pattern similarity difference between pairs of trials sampling similar (angular differences ≤ 30°) and dissimilar (angular differences ≥ 60°) directions after excluding the respective comparisons. Pattern similarity remained increased for similar directions after excluding comparisons sampling from the same start location (T_23_ = 3.01, p = 0.006) as well as after excluding comparisons targeting the same location (T_23_ = 4.07, p < 0.001). These results, which are shown in the new Figure 2—figure supplement 2, rule out the possibility that coding for locations in Donderstown underlies the observed absolute directional coding effect.

Additionally, we demonstrate in a second set of control analyses that the effect is independent of the distance between start and target location. For consistency, we turned towards the approach employed when ruling out potential distance effects in the grid-like signal observed in entorhinal cortex. This two-step procedure is mathematically similar to the multiple regression approach suggested by the reviewers in that variance explained by the distance predictor is removed before computing our effect. Specifically, we set out from the locations of the buildings entering each pair- wise comparison and quantified three distance measures for each trial pair: (I) the mean length of the vectors connecting the respective start and target buildings, (II) the difference in length between these vectors and (III) the average distance of the six vectors marking all possible connections between the four buildings of a trial pair. Calculating the same contrast as for the pattern similarity analysis (computing the difference between the mean distances for trial pairs with angular differences ≤ 30° and for pairs with larger angular differences and then testing these difference scores against 0) revealed that both the mean distance (T_23_ = 8.30, p < 0.001) and the difference in distance (T_23_ = 58.23, p < 0.001) was greater for trial pairs sampling similar directions, while the ‘neighborhood distance’ was smaller (T_23_ = -77.51, p < 0.001) for these comparisons. Crucially, absolute directional coding in the PHG was independent of all distance measures: For each subject, we used the different distance scores as regressors in three separate GLMs to predict pattern similarity values of all trial pairs in the peak voxel of the main absolute directional coding effect (Figure 2) and computed the difference in pattern similarity between pairs sampling similar versus dissimilar directions on the residuals of these GLMs. We still observed increased pattern similarity for trial pairs sampling similar directions compared to pairs sampling dissimilar directions after controlling for the mean trial distance (T_23_ = 3.90, p < 0.001), the difference in distance within a trial pair (T_23_ = 3.61, p < 0.001) and the average distance between all buildings of a pair (T_23_ = 3.97, p < 0.001). These results are shown in new Figure 2—figure supplement 2.

Taken together, the results of these control analyses support the interpretation of our effect being specific to an absolute directional code independent of locations and distances in Donderstown. The sections of the manuscript describing these analyses are shown below.

Results:

“Further, this effect was not driven by the specific locations used to sample directions or the distances between these locations in Donderstown (Figure 2—figure supplement 2, see Materials and methods).”

Materials and methods:

“Exclusion of trial pairs with the same start location. To control for the possibility that increased pattern similarity was due to pairs of trials sampling directions from the same start location, we excluded these comparisons from the analysis. […] This rules out the possibility that sampling in similar parts of Donderstown is responsible for increased pattern similarity for trial pairs sampling similar directions.”

*The authors show a grid-like representation during imagined directions, such that pattern similarity was higher when comparing trials 60° apart than 30° apart. The authors perform various control analyses to confirm the robustness of this result. The analysis ruling out the effect of cardinal directions is not reported in the main text. In this analysis they exclude trials that sample the north-south axis of the environment, doing this makes their main effect non-significant. They then use a different exclusion criterion, which leaves the main effect significant. First of all, the difference in exclusion criteria is not clear. It would be useful if the authors clarified this point. Moreover, given how much text is dedicated to this result in the method it should be included in the main text. Permitting the method that maintains the significance of their main effect is reasonable and fair, they could just report that result. Finally, it would be useful to clarify why so many trials seem to sample the north-south axis, as the author suggest lack of statistical power is what makes their main effect non-significant when you exclude these trials.*

We would like to thank the reviewers for giving us the opportunity to provide clarification on these issues. We apologize for having overlooked to refer to the analyses ruling out an effect of representations of cardinal directions in the entorhinal cortex. We have added a reference to the revised manuscript (Results, last paragraph).

The analyses in question go back to comments by a reviewer during peer review of this manuscript at a different journal. The reviewer voiced the concern that a number of roughly parallel streets running from North to South might have dominated participants’ representations of the city. Based on this we were asked to exclude all comparisons of trials sampling North or South (all arrows in Figure 4). First and foremost, we would like to emphasize that all twelve directions were sampled an equal number of times. Further, if any axis of the environment would dominate participants’ representations of the city, then this should be visible in increased/decreased pattern similarity for trial comparisons with an angular difference of 0° and/ or 180°. However, a repeated measures ANOVA did not reveal pattern similarity differences between trial pairs with different angular distances (0°, 60°, 120° and 180°) in the 0° modulo 60° condition (F_3,69_ = 1.28, p = 0.289).

Author response image 1.Trials sampling North or South are indicated by arrows.Originally, we were asked to exclude comparisons of all shown trials, leading to a reduction in statistical power. To rule out an effect of sampling along the North-South axis using buildings located on the same street we excluded comparisons indicated in purple from the analyses.**DOI:**
http://dx.doi.org/10.7554/eLife.17089.024

Additionally, behavioral performance in trials sampling the North-South axis did not differ from performance when sampling other directions (see text from manuscript below for details). Importantly, the population response of entorhinal grid cells would still remain the most likely explanation for the observed 60° modulation of pattern similarity values even if one axis was used as reference by our participants. Therefore, we believe that the non-significant result when contrasting the 0° modulo 60° condition with the 30° modulo 60° condition after excluding all comparisons along the North-South axis reflects reduced statistical power in our analysis due to the exclusion of a substantial number of comparisons. Since we believe, for the reasons laid out above, that the analysis does not add substantial information to the report, we have removed it from the revised manuscript as suggested by the reviewers.

We addressed the concern of the aforementioned reviewer that a set of streets running from North to South might drive the effect in an additional analysis. Here, we excluded comparisons of pairs sampling the North or South direction using buildings actually located on the same North-South running street (purple arrows in the figure above). We have highlighted these trials in Figure 1—figure supplement 2 of the revised manuscript to present this information more clearly to the reader. Notably, only a small subset of trials used buildings actually located on the streets in question. After excluding these trial pairs from the analysis, we observed greater pattern similarity in the peak voxel of the cluster from our main analysis (Figure 2) for trial comparisons sampling similar compared to dissimilar directions (T_23_ = 4.00, p < 0.001). Similarly, pattern similarity in left pmEC remained increased for the 0° modulo 60° compared to the 30° modulo 60° condition (T_23_ = 2.40, p = 0.025).

The adapted figure and the revised sections of the manuscript are shown below.

Results:

“Furthermore, the effect was specific to a 60° modulation of pattern similarity values and there was no evidence for coding of cardinal directions in the entorhinal cortex (Figure 3—figure supplement 9; see also Materials and methods).

Materials and methods:

“To ascertain that trial pairs sampling directions using buildings located on the same street running along the North-South axis of Donderstown (based on its orientation for example in Figure 1), which could possibly bias participants’ representations of the city due to a number of streets roughly aligned with it, were not driving our effects, we excluded all comparisons of pairs in which both trials sampled directions along this street from the analysis (see Figure 1—figure supplement 2). […] Importantly, even if participants preferentially represented directions along the North-South axis, which is unlikely based on the reasons laid out above, the population response of entorhinal grid cells would still remain the most likely explanation for the observed 60° modulation of pattern similarity values in pmEC, which could not be explained by the presence of one cardinal axis.”

*The new technique is interesting in that it removes the requirement to estimate the grid angle, but as far as we can see, it does not remove it completely because the sensitivity of the technique depends on the subject-by-subject grid angle (if the grid angle is 15° or 45° there is no sensitivity at all, but if the grid angle is 0 or 30 the sensitivity is maximal). This is due to the minimal sampling of directions. Essentially, then, the technique is similar to fitting one of sin(6θ) or cos(6θ) and ignoring the other one and taking the absolute values of the betas. Obviously there is a sensitivity gain from pooling over voxels, which may regain much of the sensitivity loss. This is obviously ok, if you find a significant effect, but these issues are not well presented in the manuscript. A naive reader might easily think that there is some magic whereby there is no longer any requirement to know the grid angle. This should be clarified.*

We would like to thank the reviewers for raising this important issue concerning our multivariate approach to investigate grid-like signals using fMRI. Contrary to the approach established by Doeller et al. (Nature, 2010) and subsequently used by Kunz et al. (Science, 2015), Horner et al. (Current Biology, 2016) and Constantinescu et al. (Science, 2016) the approach we employ does not rely on estimation of the grid orientation, but it capitalizes exclusively on the six-fold-symmetric firing patterns of grid cells. In the original paper, grid orientations were estimated per voxel and then averaged across the voxels in the entorhinal cortex. Grid orientations were significantly clustered in most participants (Doeller et al., Nature, 2010, Suppl. Figure 9). These findings are in line with recent reports of clustering of grid orientations across cells in simple environments (Krupic et al., Nature, 2015; Stensola et al., Nature, 2015). Taking the assumption of a shared grid orientation across voxels to the extreme, it is indeed conceivable that there might be no sensitivity to the 60° modulation of pattern similarity values targeted by our multivariate approach if the grid orientation is 15° or 45° with the sampling of only twelve directions.

As already alluded to by the reviewers, it can be assumed that even in the case of an average grid orientation sub-optimally suited for the multivariate approach, there would be important information carried by the variance in preferred orientations across voxels, which the representational similarity analysis approach could pick up. This stands in contrast to the voxel- wise, univariate analysis employed by Doeller et al. (Nature, 2010). Differences in grid orientation across voxels could potentially also reflect variation in the grid orientation across grid cell modules reported by Stensola et al. (Nature, 2012, Figure 2). Variation in the orientation of grid modules is also present in the two more recent studies reporting an alignment of grid orientations to environmental boundaries (Krupic et al., Nature, 2015, Figure 2; Stensola et al., Nature, 2015, Extended Data Figure 4). Differences in orientation between grid modules could result in differences in putative grid orientation across voxels in the entorhinal cortex, which might normalize sensitivity differences.

However, even when considering the possibility of orientation-differences across grid modules resulting in different preferred orientations for each voxel in the entorhinal cortex, it seems plausible to expect an overall clustering of voxel orientations at different angles for different participants in the entorhinal cortex in a complex large-scale environment such as Donderstown. This was also observed by Doeller et al. (Nature, 2010), where participants navigated an environment with multiple prominent landmarks for orientation. Potentially, this might still result in different sensitivity of our analysis across participants due to the sampling of twelve directions. While the sampling of twelve directions constitutes the minimum number of directions necessary for our analysis, a higher number of directions would have been infeasible to assess in a controlled fashion in our direction imagination task. Since the more fine-grained sampling of directions needed to estimate individual grid orientations could not be realized with our design only the multivariate approach allowed for the investigation of grid-like entorhinal signals.

We have added the discussion of this issue to the revised manuscript. Further we have modified Figure 3—figure supplement 1 for illustration. The changed sections of the manuscript can be found below.

Discussion:

“In particular, we demonstrate that this novel analysis approach, which does not rely on the estimation of the orientation of the hexadirectional signal in entorhinal cortex in an independent data set (Doeller et al., 2010; Horner et al., 2016; Kunz et al., 2015; Constantinescu et al., 2016), is sensitive to grid-like entorhinal signals by capitalizing on the six-fold symmetry of grid cell firing patterns. […] However, only the multivariate approach enabled us to investigate the six-fold rotational symmetry in our large-scale environment, in which a continuous sampling of directions as required for the estimation of the orientation of the hexadirectional signal would not have been feasible.”

*Related to the comment above, the technique presumably makes it harder to examine individual differences as the sensitivity of the technique is so dependent on the grid angle, which will differ between subjects. Is this one potential reason why the authors do not report brain-behaviour correlations? Again, the authors should discuss.*

We would like to thank the reviewers for this insightful comment. We agree with the reviewers that the potentially different sensitivity of the multivariate approach to investigate grid-like entorhinal signals discussed has to be kept in mind when considering correlations with behavior. Such a correlation is less likely to be observed, when variance in the measured strength of the neural signal across participants can be due to sensitivity differences of the analysis employed. However, as mentioned above, a continuous sampling of directions as required for the estimation of putative grid orientations was infeasible with our direction imagination paradigm.

We have added the caveat about relating grid-like signals obtained using the multivariate approach to behavior to the Discussion.

Discussion:

“This needs to be taken into consideration when aiming to relate grid-like signals to behavior.”

*In Figure 3—figure supplement 7, the authors write "We considered the distances between start and target locations in a trial pair by controlling for three distance measures, which differed between the 0 modulo 60 and the 30 modulo 60 condition". In what way? Were the 0 modulo 60 distances longer than 30 modulo 60? It is important to document this even if you can still find the hexagonal results after working with the residuals.*

We agree with the reviewers that it is important to document more clearly the differences between the conditions. The nature of the differences between the conditions was quantified separately for each distance measure. To test for differences between the conditions, we obtained a distance matrix quantifying the distance measure for each pairwise comparison of two trials, thus resembling the pairwise correlation matrix in our representational similarity analysis (RSA). Analogous to the RSA approach we computed the difference between the mean distance in the 0° modulo 60° condition and the mean distance in the 30° modulo 60° condition for each participant and tested the resulting differences against 0 using a one-sample t-test. The results of these tests, which were already reported in the manuscript, revealed that the mean trial distance (average length of the vectors from start to target building in a trial pair) was larger for the 0° modulo 60° condition (T_23_ = 5.25, p < 0.001). Further, the difference in distance between start and target building was smaller for trial pairs in the 0° modulo 60° condition (T_23_ = -57.02, p < 0.001). Lastly, the ‘neighborhood distance’ (average length of the six vectors connecting the four buildings in a trial pair), which we used as a measure of imagining directions in a certain area of Donderstown, was larger in the 0° modulo 60° condition (T_23_ = 54.20, p < 0.001). However, in light of the described differences between the 0° modulo 60° and the 30° modulo 60° condition, we would like to emphasize again that the increased pattern similarity in the 0° modulo 60° condition was not due to these differences. This effect was present after statistically accounting for variance explained by the distance measures.

We have clarified how we tested for distance differences between the conditions in the revised manuscript and documented the nature of these differences. The relevant sections of the manuscript read as follows:

Materials and methods:

“In a second set of control analyses we additionally considered the Euclidean distance between the start and the target location as described above for the absolute directional coding effect. […] The results of these tests are reported in the following paragraphs. In each control analysis, we then computed a GLM with a continuous distance predictor and pairwise Fisher z-transformed correlation coefficients as the criterion.”

Materials and methods:

“The mean Euclidean distance was higher for pairs in the 0° modulo 60° condition (T_23_ = 5.25, p < 0.001).”

Materials and methods:

“Trial pairs from the 0° modulo 60° condition had more similar distances (T_23_ = -57.02, p < 0.001).”

Materials and methods:

“This resulted in a ‘neighborhood’ distance, which was low when both start and both target buildings were in the same area of Donderstown. These distances were larger for pairs in the 0° modulo 60° condition (T_23_ = 54.20, p < 0.001).”

*More broadly, the angle selection procedure is a very clever way to increase the sensitivity of the experiment, but it raises concerns that there might be differences on average between the 30° differences and the 60° differences. Is there a way of alleviating these concerns by graphing the differences according to various metrics such as how often they include the inner vs outer ring? Such as average behavioural performance included in the two different types of trial?*

We agree with the reviewers that it is of great importance to ensure that our effects are not driven by differences between trial pairs in the 0° modulo 60° and the 30° modulo 60° condition introduced by the specific sampling of directions in our design. As suggested by the reviewers we investigated participants’ behavioral performance in more detail. Specifically, we combined angular errors for all pairwise comparisons by multiplying the two error values of each pair (e.g. a pair with pointing errors of 30° and 50° would have a value of 1500 in this analysis) and averaged the combined error values for the two conditions. We tested for a difference in behavioral performance using a one-sample t-test on the difference between the conditions. Behavioral performance did not differ between the 0° modulo 60° and the 30° modulo 60° condition (T_23_ = 1.24, p = 0.227). We have highlighted this analysis in the manuscript and added Figure 3—figure supplement 8 to illustrate equal behavioral performance in the two conditions.

Further, we followed the reviewers’ suggestion to focus on trials, where directions were sampled using a target location in the inner ring of buildings. Pairwise comparisons in which both trials sampled a direction using an inner ring target location made up 12.23% of comparisons in the 0° modulo 60° condition and did not occur in the 30° modulo 60° condition (see Figure 5). Since we observed lower pointing errors for trials with targets in the inner ring compared to the outer ring (one-sample t-test on the difference between errors, T_23_ = -5.29, p < 0.001), we conducted an additional control analysis to exclude the possibility that the increase in pattern similarity in the 0° modulo 60° condition was due to comparisons of trials with targets in the inner ring. In this control analysis, we excluded pairwise comparisons in which both trials used a target building located in the inner ring before computing the pattern similarity difference between the 0° modulo 60° condition and the 30° modulo 60° condition in each subject’s left pmEC ROI. Pattern similarity remained increased for the 0° modulo 60° condition compared to the 30° modulo 60° condition (T_23_ = 5.29, p < 0.001). This rules out that our effect is due to an over-representation of trials targeting buildings in the inner ring.

Author response image 2.Percentage of pairwise comparisons in the two conditions as a function of the number of trials in a pair targeting a building located on the inner ring.Trial pairs in which both trials targeted a building in the inner ring made up 12.23% of comparisons in the 0° modulo 60° condition and did not exist in the 30° modulo 60° condition.**DOI:**
http://dx.doi.org/10.7554/eLife.17089.025

We have added this control analysis to the manuscript and show its results in Figure 3—figure supplement 6. The revised sections of the manuscript now read as follows.

Results:

“Behavioral performance did not differ between the conditions (T_23_ = 1.24, p = 0.227, Figure 3—figure supplement 8).”

Materials and methods:

“We investigated whether a 60° modulation of absolute angular error values was also present in the behavioral data. […] The combined error values were not different between the conditions (T_23_ = 1.24, p = 0.227, Figure 3—figure supplement 8).”

Materials and methods:

*“*Exclusion of trial pairs with target locations in inner ring of buildings. In some trials, directions were sampled using target buildings located on the inner ring of buildings (see Figure 1—figure supplement 2). […] When excluding these comparisons from the analysis, pattern similarity remained increased for the 0° modulo 60° condition compared to the 30° modulo 60° condition (Figure 3—figure supplement 6,bar V).”

*There is also a concern with pattern similarity techniques of autocorrelation. We have thought about it but the various combinations are so mind-bendingly complex that it is hard to know whether this will be true. Are there some trials that are more often included in the 60° pairs than the 30° pairs? If there are more or less of these than their complements (trials that occur mostly in the 30° pairs), is it possible that they can cause some statistical autocorrelation in the question of interest. We think this is unlikely, but would be interested to hear your thoughts. We are aware that this is a weakly specified criticism. If no analyses come to mind, some comment to the reviewers would suffice.*

We understand the concern raised by the reviewers as representational similarity analysis is very sensitive to temporal autocorrelation. However, we do not see how autocorrelation could drive the reported effect. Since the analysis is conducted based on the relative angular difference between the homogenously sampled directions, each trial is included equally often in both conditions.

Therefore, the only possibility for autocorrelation to contribute to the effect would be if trials in one condition were presented closer together in time. However, we do not believe this is the case for two reasons: One of the control analyses reported in the manuscript showed that pattern similarity was increased in the 0° modulo 60° condition when only comparing voxel patterns of trials recorded in different scanning runs (T_23_ = 2.08, p = 0.049; Figure 3—figure supplement 6, bar IV). We believe that excluding comparisons of trials from the same run makes a contribution of temporal autocorrelation to the effect highly unlikely as the scanner was stopped in between runs and runs were separated by breaks typically lasting a few minutes to allow participants to rest. These breaks were terminated by the participant indicating he was ready to continue with the next task block.

Further, we followed up on the comment by the reviewers by assessing the time between trials constituting the pairs in both conditions. We quantified the time between the onsets of the imagination periods of the two trials of each pair for all pairwise comparisons within a run. There was no apparent difference between the 0° modulo 60° condition and the 30° modulo 60° condition.

Author response image 3.Mean proportion of pairwise comparisons in the 0° modulo 60° condition and the 30° modulo 60° condition across participants for different time bins.The onset asynchrony was defined as the absolute difference in seconds between the onsets of the imagination periods of the trials in a pair. Error bars reflect SEM.**DOI:**
http://dx.doi.org/10.7554/eLife.17089.026

Taken together, we believe that these data speak against the possibility that our effect might be influenced by autocorrelation.